# Resilience profiles across context: A latent profile analysis in a German, Greek, and Swiss sample of adolescents

Clarissa Janousch[1]*, Frederick Anyan[2], Wassilis Kassis[1], Roxanna Morote[2,3], Odin Hjemdal[2], Petra Sidler[1‡], Ulrike Graf[4‡], Christian Rietz[4‡], Raia Chouvati[5‡], Christos Govaris[5‡]

1 Institute for Research and Development, School of Education, University of Applied Sciences and Arts Northwestern Switzerland, Brugg-Windisch, Switzerland, 2 Department of Psychology, Norwegian University of Science and Technology, Trondheim, Norway, 3 Department of Psychology, Catholic University of Peru, Lima, Peru, 4 Faculty of Educational and Social Sciences, University of Education Heidelberg, Heidelberg, Germany, 5 Department of Primary Education, University of Thessaly, Volos, Greece

☯ These authors contributed equally to this work.
‡ PS, UG, CR, RC and CG also contributed equally to this work.
* clarissa.janousch@fhnw.ch

**Data Availability Statement:** All relevant data are within the manuscript and its Supporting Information files.

## Abstract

The present study investigated resilience profiles (based on levels of symptoms of anxiety and depression and five dimensions of protective factors) of 1,160 students from Germany ($n = 346$, 46.0% females, $M_{age} = 12.77$, $SD_{age} = 0.78$), Greece ($n = 439$, 54.5% females, $M_{age} = 12.68$, $SD_{age} = 0.69$), and Switzerland ($n = 375$, 44.5% females, $M_{age} = 12.29$, $SD_{age} = 0.88$) using latent profile analyses. We also checked for measurement invariance and investigated the influence of gender and migration on class membership. A three-profile-solution was found for Switzerland (nonresilient 22.1%, moderately resilient 42.9%, untroubled 34.9%), and a four-profile-solution was the best fitting model for Germany (nonresilient 15.7%, moderately resilient 44.2%, untroubled 27.3%, resilient 12.7%) and Greece (nonresilient 21.0%, moderately resilient 30.8%, untroubled 24.9%, resilient 23.3%). Measurement invariance did not hold across the three countries. Profile differences regarding class membership predictions were detected for Germany and Greece, but none for Switzerland. Results implicate that resilience profiles are highly contextually sensitive, and resilience research findings should not be generalized considering the particularity of contexts, people, and outcomes.

## 1 Introduction

Instead of only understanding resilience as a linear set of causal relationships, recent research on resilience has focused on the multisystemic aspect of the concept [1]. Resilience of human and ecological systems are mutually dependent on each other [2] and therefore, resilience needs to be studied by taking the different contexts of these systems and their connectivity into account. Furthermore, resilience is a normative concept that is highly influenced by cultural aspects such as moral values, and structural and social dimensions [3]. Ungar and Theron

**Funding:** This study was funded by the Swiss National Science Foundation (SNSF) through the National Centre of Competence in Research (NCCR) – on the move via the project Overcoming Inequalities with Education – School Resilience, grant number 51NF40-182897, awarded to WK. Additionally, the University of Applied Sciences and Arts Northwestern Switzerland (FHNW) generously supported the study. The funders had no role in study design, data collection and analysis, decision to publish, or preparation of the manuscript.

**Competing interests:** The authors have declared that no competing interests exist.

raised the question, "Which promotive and protective factors or processes are best for which people in which contexts at what level of risk exposure and for which outcomes?" [4]. This paper represents an attempt to contribute to answering this question by conducting a latent profile analysis (LPA) across three independent samples using the level of symptoms of anxiety and depression, and five subdimensions of protective factors–namely personal competence, social competence, structured style, family cohesion, and social resources–showing how highly contextually influenced resilience profiles are.

As a transitional stage from childhood to adulthood, adolescence is considered a crucial phase of major physical, cognitive, and socio-affective changes that may lead to psychological and physiological growth [5]. However, certain life events and risk factors can affect adolescents' well-being. According to the World Health Organization (WHO; [6]), mental disorders such as depression and anxiety commonly emerge during this period. Between the ages of 15 and 19 years, depression is the fourth and anxiety the ninth leading cause of illness and disability for adolescents worldwide, and they are often comorbid [7]. Anxiety and depression comorbidity can cause lower life satisfaction, more physical and mental health problems, suicide, academic difficulties, and greater overall impairment [8].

Results from the cross-cultural KIDSCREEN study [9] that examined emotional well-being in a representative sample of 22,000 children and adolescents in 12 European countries accentuate the need for cross-cultural comparisons and differentiations when discussing mental health problems. Even though it is a global issue, the prevalence of depressive and anxiety symptoms varies highly across cultures. On one hand, Germany ($n$ = 1,113) and Switzerland ($n$ = 1,121) reached the lowest prevalence rates among all 12 participating countries. The authors classified the adolescents' test results into so-called "normal," "borderline," and "abnormal" mental health, resulting in 7.1% borderline and 2.9% abnormal in Germany with 6.4% borderline and 3.6% abnormal in Switzerland. On the other hand, Greece ($n$ = 1,174) was among the highest rates, with 13.8% borderline and 7.2% abnormal. These findings can be explained by several factors, such as the families' respective socioeconomic situation (which might be influenced by the economic situation of a country or region) that has proven to be associated via perceived social status with adolescents' mental health [10]. Additionally, structural aspects, such as parental mental health issues, domestic violence, or poor peer support can influence adolescents' mental health [9, 11]. All these aspects are unique but closely linked factors contributing to adolescents' mental health and resilience. However, the extent of influence of each factor differs due to cultural features in a society.

Migration background also has a critical impact on adolescents' mental health because it is influenced by social factors on personal, family, community, and national levels [12]. Several studies and reviews have shown that migration in Europe is associated with an elevated risk of emotional problems and greater psychological distress in adolescents and adults [e.g., 13, 14]. Adverse effects of an ethnic minority status (e.g., experience of exclusion and/or discrimination), the migration process (e.g., forced migration, loss of cultural connectedness, the use of another language, and acculturation), and an inferior socioeconomic status can explain the poorer mental health of migrant adolescents [15–17].

Finally, the KIDSCREEN study also detected group-specific variations related to gender: More girls were among the two risk groups (with the exception of Greek boys in the "abnormal" profile; 8.2% vs. 6.5%; [9]). Nevertheless, it has been well established that girls are at greater risk for mental health problems than boys due to the interaction between biological factors such as hormones and determinants on a societal level, including gender stereotypes, lack of gender equality, and gender-based discrimination [18–21]. In particular, girls are about twice as likely to be affected by depressive symptoms and diagnosable disorders than boys [22, 23].

However, it is essential to recognize that a notable number of adolescents do not develop mental health or behavioral disorders. This is supported by general findings from the KIDSC-REEN study; 90% of adolescents in Germany and Switzerland and almost 80% of the Greek sample were classified among the normal mental health group (in terms of anxiety and depression levels). Yet, even when exposed to risk, a substantial number deal adequately with risk factors [24]. This is when studying protective factors becomes key. Literature has shown that several relational protective factors influence positive outcomes in multiple domains, including the individual, family, and societal levels [25, 26]. According to Rutter [27, 28], protective factors are individual, contextual, or external conditions that help resist or balance the negative impacts of the risks to which a person is exposed. Similarly, it appears that there are gender-specific differences regarding protective factors of mental health. The most common internal attributes for adolescents at risk for developing mental health issues such as anxiety and depression include perseverance, self-efficacy, creativity, self-awareness, self-regulation, self-esteem, optimism, and active coping strategies [e.g., 26, 29–31]. Girls score higher in social competencies; boys express more self-confidence and self-esteem [32–35]. On the family level, support, positive feedback, cohesion, and good interaction with parents are vital for both genders [29, 36–39]. However, the literature suggests that family cohesion buffers the effects of stress for girls more than for boys [40, 41]. Other external protective factors for girls and boys are good interpersonal relationships to not only family members but also peers [36], supportive teachers [38], and good community resources [42–44]. A recent study concluded that the compensatory effect of protective factors against depressive symptoms is particularly strong for girls [45].

Research has shown specific protective factors related to individuals having a migration background: Social resources (having a feeling of connectivity, belonging, and good relationships) are the most beneficial protective factors to preserve and foster the mental health of people with migration backgrounds [46]. Thus, the possible lack of community belonging, a feeling of isolation, and/or social needs not being met can increase the chance of mental health problems [47, 48].

A methodological review of resilience scales has already shown in 2011 [49] that measuring resilience is challenging due to its ambiguous definition. Most of the time, resilience is being measured as the presence or absence of assets and resources that facilitate resilience as a process, but no "gold standard" was found among the 15 measures. Even though, resilience scales mostly cover only assets and resources, models and concepts of resilience go beyond analyzing these aspects and include risks factors, often focusing on the absence of negative indicators of mental health (e.g., anxiety or depression). A meta-analysis of 31,071 participants in 33 studies investigated the relationship between psychological resilience and relevant variables [50]. All selected articles stem from the years 2001 to 2010 and results indicated as expected that protective factors, such as self-efficacy, life satisfaction, or optimism have the biggest effect on resilience. In addition, medium effects were measured for risk factors, such as depression, anxiety, or PTSD. Finally, demographic variables contributed small effects, but were still important to resilience. Another meta-analysis supported these findings by analyzing the relation between resilience, mental health, and demographics in 60 studies representing 68,720 participants [51]. High correlations were found between resilience and mental health. Additionally, gender was moderating this relationship. More attention needs to be paid to females experiencing higher levels of mental health problems and lower levels of protective factors. Therefore, when investigating resilience, it is crucial to not only consider the protective and risk factors but also to include mental health aspects and demographics in the research questions.

For several decades, resilience scientists have been on a prolonged mission to understand mental health issues to prevent and treat them by examining risk and protective factors. Instead of focusing on pathways leading toward psychopathology in pathogenesis, resilience research arose from attempts to account for both positive and negative patterns [52] based on

a salutogenic approach to health [53]. Despite definitional ambiguity, newer definitions reflect the perspective of resilience as a complex, dynamic, and adaptive system that goes beyond the idea of an individual bouncing back and recovering from a traumatic experience. For the purpose of this paper, we define resilience according to Masten as "the capacity of a dynamic system to adapt successfully to disturbances that threaten systemic function, viability, or development" [30]. This definition does not only accentuates the multisystemic nature of resilience but subsequently acknowledges the importance of cultural narratives and contextual realities in mental health and resilience research [1, 4, 54, 55]. This conceptualization is based on the idea of Bronfenbrenner's ecological systems theory [56, 57], stating that development always emerges from an individual's interaction within a context as part of multiple systems, such as the family, a peer group, or a neighborhood, influencing them while being influenced by them. Accordingly, risk, protective factors and resilience outcomes might neither be similar nor comparable across groups or contexts [58, 59]. Consequently, researchers must consider these influences on mental health and resilience on all levels, namely the individual level, family support, and support by the broader environment. Therefore, this complexity of socioecological resilience leads, according to Ungar and Theron [4], to investigating resilience factors considering the particularity of contexts, people, the levels of exposure, and possible outcomes.

## 1.1 Person-centered approaches in resilience research

The difficulty of defining resilience leads to ambiguity in operationalizing the concept. Masten frequently refers to two methodological approaches of resilience research—the variable-centered and the person-centered approach [30, 60–63]. The variable-centered approach (e.g., correlation, regression, structural equation modeling, factor analysis) is the traditional and dominant approach in social sciences to analyze associations among variables of interest. It investigates research questions regarding one variable's effects on another by collecting data from many subjects for at least one occasion. Even though this approach has statistical power and can generate general patterns between variables, it cannot identify dynamics of emergent subpopulations in a sample [64]. This is where the person-centered approach comes into play. It investigates research questions regarding categorizations of subjects into common subpopulations, based on a set of relevant and chosen variables (e.g., latent class analysis, LPA, and latent transition analysis). First, this approach follows the notion that there is no homogeneity within a population; thus, distinct subgroups may exist within this population, as individuals are part of a totality rather than a summation of variables. Second, these subgroups (so-called profiles or classes), if they exist, differ qualitatively as well as quantitatively. Individuals in one group are similar to each other but differ from those in other groups, and therefore, predictors, correlates, or outcomes can be closely analyzed [64–67].

As Masten [63] stated, person-focused models study individuals as a whole unit of interest, making it possible to identify resilient and nonresilient individuals and to compare them to each other. These models are well suited to search for genuine profiles that occur in real people because researchers cannot define "true" adversity and risk factors for every individual. Even though in samples with similar risk factors (e.g., homelessness), striking variability can be found; thus, risk and its perception are group-dependent [68]. Therefore, instead of classifying a dichotomy or bipolar dimension of normal and abnormal, healthy and sick, or resilient and nonresilient, person-centered approaches show individuals with different combinations of strengths and weaknesses and, thus, a variety of resilience outcomes.

There is an increasing number of empirical studies examining profiles (aspects) of resilience using latent class and profile analyses (LCA & LPA). A recent study from the United Kingdom

and Western Australia, for instance, focused mainly on adversity (i.e., different configurations of lifetime adversities) and resilience resources (i.e., bounce-back, hope, self-efficacy, and optimism) [69]. The team used the adapted version of the cumulative lifetime adversity measure [70, 71], the Bounce-back ability [72], the Adult Hope [73], the General Self-Efficacy [74] / a self-efficacy scale by Bell and Kozlowski [75], and the Life Orientation Test-Revised Scale [76]. They conducted two separate studies with a general ($N$ = 1,506, 48.2% females) and a university sample ($N$ = 348, 61.5% females). Results revealed three profiles for each sample showing statistically different levels of resilience in the three detected classes. For the general sample, the biggest group was the Moderate class (62.7%) followed by the High (20.5%) and Low (16.8%) Polyadversity classes. In the university sample, the Low (41.1%) and High Polyadversity (41.1%) classes showed identical group sizes, while the Vicarious Adversity (17.8%) was the smallest group. Differences between all three latent classes in both subsamples in terms of individual-level resilience resources were mixed. Individuals in the Moderate Polyadversity class reported the highest level of resilience in the general population study. These findings were statistically significant when comparing the Moderate class with the High Polyadversity class, but only partially significant (for bounce-back resilience and optimism) in comparing the Moderate with the Low Polyadversity classes. According to Lines et al. [69] and previous studies [77, 78], a moderate amount of exposure to adversity is ideal for opportunities to develop protective factors, and therefore to support resilience.

Furthermore, being less exposed to adversity led to more resilience resources (protective factors) when comparing the Low and High Polyadversity classes. Being exposed to a high amount of adversity is highly detrimental regarding the availability of protective factors. Additionally, being exposed to fewer adversities might give fewer opportunities to develop necessary resilience resources compared to a moderate amount of adversities. Additionally, gender differences were detected in both samples. Females were more likely than males to be part of the High Polyadversity class than the Vicarious Adversity or Low Polyadversity classes in the university sample, while males were more likely to be part of the Low Polyadversity class compared to the Moderate Polyadversity class in the general sample. No information was given on migration background.

Another study focusing on mental health classifications examined profiles of American high school students ($N$ = 332, 48.5% females) over 3 years using a dual-factor construct of mental health [79]. Measurements used included the Social-Emotional Health Survey-Secondary [80] and the Strengths and Difficulties Questionnaire [81]. Independent LPAs for each grade [9–12] based on four positive mental health domains and internalizing and externalizing problems revealed four distinct subgroups—Complete Mental Health, Moderately Mentally Healthy, Symptomatic but Content, and Troubled. Like the general population sample in the study mentioned above [69], most students were in the Complete (30.5% Grade 9, 40.8% Grade 10, 20.5% Grade 11) or Moderately Mentally Healthy (43.4%, 32.0%, 44.3%) classes. The Troubled class (5.7%, 6.0%, 3.8%) represented the smallest number of individuals across all grades, while the Symptomatic but Content class (20.3%, 21.2%, 31.3%) was between these classes. Higher levels of distress and lower levels of strength were reportedly associated with fewer symptoms of anxiety and depression. No further investigations on gender and migration background were made.

This four-profile-solution was confirmed by Reinhardt et al. [82] based on three well-being indicators (emotional, psychological, and social aspects), resulting in the Languishing, Moderate Mental Health, Emotionally Vulnerable, and Flourishing classes. 1,572 (51% females) Hungarian adolescents filled out a questionnaire including the Adolescent Mental Health Continuum–Short Form [83] and the Strength and Difficulties Questionnaire [81]. 39% were part of the Moderate Mental Health group, 11% belonged to the Emotionally Vulnerable class. The Languishing class, including 14% of the sample, reported low levels of prosocial behavior, high rates of peer problems, and loneliness. In contrast, lower levels of loneliness, more

prosocial behavior, and fewer emotional problems and peer problems predicted the Flourishing class (36%) in comparison to the Languishing class. Furthermore, the Flourishing category included more males and younger adolescents compared to the Languishing group. No further gender differences were reported, nor was any information on migration background.

Two further recent studies focused on risk and protective factors. Mohanty et al. [84] were able to demonstrate that protective factors on all levels might play a crucial role in preventing the occurrence of risks in their three classes: Moderate (39.5%), Protective (34.3%), and High-risk (26.2%). In the study, 953 (67.2% females) participants answered a questionnaire including two items measuring pre-adoption risk [85], eleven self-created items about post-adoption risk [84], the Multidimensional Scale of Perceived Social Support [86], and a single item asking "how many close friends do you have?". Findings suggest that social support in particular ameliorated negative effects of risks. More males were part of the Moderate class, while more females were part of the Protective and High-risk classes. Migration background has not been investigated.

Finally, a four-class-solution is supported by Altena et al. [87], confirming that accumulated protective factors are important in preserving a certain quality of life. Findings resulted in the four classes, High-Risk and Least Protected (24%), Higher Functioning and Protected (14%), At-Risk (45%), and Low-Risk (17%) classes. 251 adolescents (32% females) participated in the study that were asked as a single item whether they have been abused. Furthermore, they answered questions from the Lehman Quality of Life Interview [88], the Brief Symptom Inventory-53 [89], the European Addiction Severity Index [90], the Resilience Scale [91] and the Cognitive Emotion Regulation Questionnaire [92]. No gender differences existed between the subgroups and migration background was not investigated in the study.

However, both studies included a very specific sample. Mohanty et al. [84] focused on Korean adult international adoptees, whereas Altena et al. [87] investigated homeless young adults in the Netherlands. Thus, findings need to be treated with caution when comparing to more general samples.

### 1.2 Present study

We used the person-centered approach to investigate Ungar and Theron's question ("Which promotive and protective factors or processes are best for which people in which contexts at what level of risk exposure and for which outcomes?"; [4] and identify dynamics of emergent subpopulations in various samples. We explored resilience profiles (symptoms and protective factors) via LPA. Furthermore, we examined whether these profiles are comparable across three country samples (a German, Greek, and Swiss sample) and how the identified profiles differ regarding gender and migration background. This leads to the following research questions and hypotheses:

1. How many resilience profiles based on symptoms (depression and anxiety) and protective factors (personal competence, social competence, structured style, social resources, and family cohesion) can be found?
   (H1) Based on previous LCA-/LPA-studies [e.g., 69, 79, 82], at least three resilience profiles were hypothesized for each country.

2. Do identical resilience profiles exist across Germany, Greece, and Switzerland?
   (H2) Following Ungar and Theron's principle [4], identical resilience profiles were not expected due to cultural sensitivity of contextual effects on resilience.

3. Are gender and migration background predictors of latent resilience profiles?
   (H3) According to several research findings [e.g., 9, 13, 14], gender and migration background increase the probability that individuals belong to a specific profile.

## 2 Materials and methods

### 2.1 Participants

This study examined data (data available in the Supporting information section, S1 File) collected as part of the National Centres of Competence in Research (NCCR)–On the Move via the project *Overcoming Inequalities with Education–School and Resilience*, which was administered in 2019. The project also investigates protective and risk factors of mental health through a web-based survey. Participants were a random sample based on convenience, consisting of seventh graders from lower secondary education classes (ISCED 2) in rural and urban regions of Germany, Greece, and Switzerland.

Three hundred forty-six students (46.0% female; 20.6% natives) from schools in Baden-Württemberg, Germany filled out the questionnaire. This sample's mean age was 12.77 ($SD$ = 0.78), ranging from 11 to 16 years. Similar to the German sample, the Swiss sample contained fewer female students. Of the 375 participants, 44.5% were female and 24.6% were natives. The whole sample ranged from 11 to 15 years, with a mean value of 12.29 ($SD$ = 0.88). The team collected data in Switzerland in the Canton of Aargau, Basel-City, and Solothurn. With the most extensive range of 11 to 20 years, the students from Greece represented the largest subsample, with 439 girls and boys. Unlike the other two subsamples, the number of females was higher (54.5%), but the mean age was between the German and Swiss subsamples ($M$ = 12.68, $SD$ = 0.69). Additionally, the Greek subsample consisted of the highest number of natives (54.0%) The research team collected this subsample's data in Athens, Crete, and Volos.

### 2.2 Procedure

The data collection in all three countries has been conducted in accordance with the World Medical Association's Declaration of Helsinki. The Ministry of Education, Youth, and Sports Baden-Württemberg, the Common Ethics Committee of the University of Education Heidelberg, and the Stiftung Rehabilitation Heidelberg (SRH; Germany); the General Assembly of the Pedagogical Department of Primary Education of the University of Thessaly (Greece); and the Cantonal Bureau for Education in the Cantons of Aargau, Basel-City, and Solothurn, as well as the Ethics Committee of the Faculty of Arts and Social Sciences of the University of Zürich (Switzerland), endorsed the data collection.

First, several schools in the mentioned regions were asked to participate in the study. After each school's principals consented, teachers, parents, and children received an information letter and a consent form explaining the procedure and stating that participation was voluntary, anonymous, and confidential. At any point, participants were able to withdraw from the study. Thus, written consent to participate in the study was provided by the students and their legal guardians.

All participating students filled out a web-based questionnaire that took no more than 90 minutes ($M$ = approx. 48 minutes, $MD$ = approx. 47 minutes). Even though teachers helped to administer the survey, research associates conducted the data collection and answered students' questions. Students who were not available during these 90 minutes were asked to fill out the survey at a later stage but as soon as possible.

### 2.3 Measures

**2.3.1 Demographics.** Participants provided data on age, gender, nationality, and country of birth of themselves and their parents. As we were interested in whether any kind of migration background influences resilience patterns, we used nationality and country of birth to operationalize students' migration background. The term "migration background" is an official statistical category that got introduced in Germany in 2005 [93]. According to the Federal

Statistical Office of Switzerland, migration background is defined by the individual and their parents' place of birth, and the individual's nationality at birth [94]. Based on these references, not having a migration background was defined as neither the students nor their parents being born in a country other than the relevant country of data collection (here: Germany, Greece, or Switzerland) or parents and/or students possessing a passport other than the corresponding one (here: German, Greek, or Swiss passport). Thus, if one of these conditions mentioned applied, the respective student was categorized as having a migration background. However, as migrant students can be heterogeneous concerning countries of origin and migration generation [95], this categorization involves loss of information. Nevertheless, analyzing the relevant resilience profiles, which already dissect the national sample sizes (range $N$ = 346–439), such as binary categorization (migration background = 0; natives = 1), allows us to exploit the data.

**2.3.2 Resilience scale for adolescents.** The Resilience Scale for Adolescents (READ; [33]) has 28 items. On a five-point Likert self-report scale, participants are asked to rate only positively phrased items (e.g., I reach my goals if I work hard), ranging from 1 (*Totally disagree*) to 5 (*Totally agree*) in five subscales (personal competence, social competence, structured style, social resources, and family cohesion). Higher mean scores on the READ indicate higher levels of protective factors. As in other (cross-cultural) studies [35, 96] the Cronbach's alpha and McDonald's omega of this valid and reliable measurement lie across all countries and subscales between alpha and omega of .58 (Structured Style, in the Swiss subsample), and $\alpha$ = .90 (READ total scale in the German and Swiss subsamples), respectively; $\omega$ = .91 (READ total scale in the Greek subsample). Please see Table 1 for detailed means, standard deviation, and reliability values of each subsample.

**2.3.3 The Hopkins symptom checklist.** Symptoms of anxiety and depression were assessed through the Hopkins Symptom Checklist (HSCL-25; [97, 98]), originally derived

**Table 1. Means, standard deviations, Cronbach's alpha, McDonald's omega.**

| | Country | $M$ | $SD$ | $\alpha$ 95% CI [LL, UL] | $\omega$ 95% CI [LL, UL] |
|---|---|---|---|---|---|
| READ | GER | 4.06 | .53 | .90 [.88, .92] | .90 [.87, .92] |
| | GRE | 4.13 | .51 | .88 [.85, .90] | .91 [.88, .96] |
| | SWI | 4.09 | .49 | .90 [.88, .92] | .90 [.87, .92] |
| PC | GER | 3.88 | .63 | .72 [.64, .78] | .72 [.62, .78] |
| | GRE | 3.93 | .61 | .75 [.70, .79] | .74 [.68, .78] |
| | SWI | 3.89 | .57 | .74 [.68, .79] | .74 [.67, .79] |
| SC | GER | 3.95 | .70 | .66 [.57, .73] | .66 [.55, .73] |
| | GRE | 4.12 | .67 | .66 [.59, .73] | .66 [.59, .72] |
| | SWI | 4.02 | .65 | .70 [.63, .75] | .70 [.63, .75] |

(*Continued*)

**Table 1.** (Continued)

| | Country | M | SD | α 95% CI [LL, UL] | ω 95% CI [LL, UL] |
|---|---|---|---|---|---|
| SS | GER | 3.61 | .82 | .60 [.51, .67] | .60 [.51, .66] |
| | GRE | 3.75 | .82 | .59 [.51, .65] | .59 [.51, .67] |
| | SWI | 3.67 | .71 | .58 [.50, .65] | .58 [.50, .66] |
| SR | GER | 4.50 | .62 | .77 [.69, .83] | .77 [.68, .83] |
| | GRE | 4.57 | .59 | .74 [.66, .80] | .74 [.66, .80] |
| | SWI | 4.50 | .59 | .79 [.73, .83] | .79 [.73, .83] |
| FC | GER | 4.32 | .69 | .82 [.77, .86] | .82 [.77, .86] |
| | GRE | 4.26 | .67 | .79 [.75, .83] | .79 [.74, .83] |
| | SWI | 4.33 | .68 | .86 [.82, .89] | .86 [.82, .90] |
| HSCL | GER | 1.86 | .61 | .94 [.92, .95] | .94 [.92, .95] |
| | GRE | 1.74 | .55 | .93 [.91, .94] | .92 [.91, .94] |
| | SWI | 1.85 | .60 | .94 [.93, .95] | .94 [.93, .95] |
| Anx | GER | 1.89 | .59 | .83 [.79, .86] | .83 [.79, .86] |
| | GRE | 1.77 | .62 | .87 [.84, .89] | .86 [.84, .88] |
| | SWI | 1.95 | .62 | .86 [.83, .88] | .86 [.82, .88] |
| Dep | GER | 1.80 | .69 | .92 [.90, .94] | .92 [.90, .94] |
| | GRE | 1.71 | .57 | .88 [.86, .90] | .88 [.86, .90] |
| | SWI | 1.78 | .68 | .93 [.91, .94] | .93 [.91, .94] |

*Note*. GER = Germany; GRE = Greece; SWI = Switzerland; READ = Resilience Scale for Adolescents; PC = personal competence; SC = social competence; SS = structured style; SR = social resources; FC = family cohesion; HSCL = Hopkins Symptom Checklist; Anx = anxiety; Dep = depression; ω = McDonald's omega.

from the 90-item Symptom Checklist (SCL; [99]). Although the original scale contains 25 items, only 24 items were used. The item "Loss of sexual interest or pleasure" was left out due to the participants' age range. The valid and reliable scale uses a four-point Likert scale ranging from 1 (*Not at all)* to 4 (*Extremely*; e.g., *Feeling fearful*). Cronbach's alpha and McDonald's

omega ranged from .83 (anxiety subscale in the German subsample) to .94 (HSCL total scale in the German and Swiss subsamples) across all countries and subscales.

## 2.4 Statistical analyses

LPA with continuous indicators of anxiety, depression, and the five subscales of the READ (personal competence, social competence, structured style, social resources, and family cohesion) was conducted in Mplus 8.3 [100] to identify the best-fitting solution for each country sample. Maximum likelihood estimation with robust standard errors (MLR) was used to assess the classification of participants. Progressively larger numbers of latent class (one-profile to five-profile) solutions were run to determine the optimal number of profiles. First, to avoid converging on suboptimal local maxima, all models were estimated with 500 random start values and 50 iterations, and the best 100 solutions were kept. In a second step, the random start values were increased to 2000 and the iterations to 500, and the 100 best solutions were retained, confirming the initial results. According to Morin et al. [101], the means and variances were freely estimated in all profiles and models. A variety of fit statistics, the substantive meaningfulness of the profiles, and their theoretical interpretability were analyzed to determine the optimal solution. The Akaike information criterion (AIC; [102, 103]), Bayesian Information Criterion (BIC; [104]), sample-size adjusted BIC (aBIC; [105]), Vuong-Lo-Mendell-Rubin Likelihood Ration test (LMR-LRT; [106]), Lo-Mendell-Rubin Adjusted Likelihood Ratio test (aLMR-LRT; [106]) and Bootstrapped Likelihood Ratio test (BLRT; [67, 107, 108]) were all examined. Entropy values were reported, showing the precision of the classification across the profiles with values ranging from 0 (lower accuracy) to 1 (higher accuracy). When comparing a K-class model with a K-1 class model, a significant LMR-LRT and an aLMR-LRT test indicated that the model with K classes is the best fitting solution. We sought a model with lower values for the criterion indices, higher entropy values, and significant $p$ values for the BLRT [109, 110]. Fit indices, in combination with theoretical interpretability, guided the final model selection. Once the final solution was identified, a Wald test was used to examine significant differences between parameters (i.e., means), and afterwards, the profiles were compared on the basis of included covariates (gender and migration background) using the Mplus R3STEP Auxiliary function to predict class membership. We investigated whether gender and migration background are related to a higher probability of participants belonging to one specific profile rather than another. The R3STEP maintains a stable class solution and less biased parameter estimates, and coefficients are interpretable for the covariates [111].

The study of measurement invariance in LPA (MI-LPA) is necessary to evaluate whether the latent profiles' number and nature are the same across certain groups (here, across the three countries' samples: Germany, Greece, and Switzerland) observed using a series of nested models [112]. The MI-LPA was tested by comparing an unconstrained model with the same number of profiles and freely estimated means to a means-constrained model across the three countries' samples. The means-constrained model should fit the data better than the unconstrained model for invariance to hold. Thus, the profiles are qualitatively and quantitatively comparable across the samples. Smaller AIC, BIC, and aBIC values, and a significant LRT test in the unconstrained model in relation to the constrained model indicate that measurement invariance does not hold. Noninvariance would mean that the profiles are characterized differently across the country samples, and therefore are not directly comparable and interpretable [112–114]. The MI-LPA was conducted for three- and four-profile model solutions (Switzerland, Germany, and Greece) in two separate analyses. A third analysis was conducted for a four-profile model solution (Germany and Greece).

# 3 Results

## 3.1 Descriptive statistics

Univariate analyses of variance revealed that the Greek sample scored significantly higher on the social competence scale than the German sample did, $F(2, 1063) = 5.98$, $p < .05$, and significantly lower than the German and Swiss samples did on the HSCL total and its anxiety subscale, $F(2, 1064) = 4.63$, $p < .05$, and $F(2, 1014) = 8.29$, $p < .001$. However, no statistically significant differences were found for any other mean values (see Table 1). These results indicated that partial differences in certain indicators of the three groups for these items could be expected.

## 3.2 Class identification of the LPA

To consider the context-dependent and nonuniversal nature of resilience, and to determine whether we could find the same number of profiles in each national sub-sample, we defined separate LPA models for the German, Greek, and Swiss samples. The model fit indices of all of the tested LPAs are detailed in Table 2.

The one-profile solution showed the highest AIC, BIC, and aBIC values in all three subsamples, which indicated the worst fit. Furthermore, significant LMR LR tests, aLMR LR tests, and BRLRT tests in the two-profile solution supported the idea of rejecting a single-profile solution in favor of at least two classes in all three subsamples.

For the Swiss subsample, a nonsignificant LMR LR test and a nonsignificant aLMR LR test indicated that a four-profile solution did not fit better than a three-profile solution did.

**Table 2. Model fit indices for latent profile analyses of Germany, Greece, and Switzerland.**

| | Number of Profiles | AIC | BIC | ABIC | Entropy | LMR LR Test p-values | ALMR LR Test p-value | Sample proportion per class | Classification accuracy | BLRT p-value |
|---|---|---|---|---|---|---|---|---|---|---|
| GER | 1 | 4503.762 | 4557.449 | 4513.038 | | | | 342 | | |
| | 2 | 3767.690 | 3878.900 | 3786.905 | .818 | < .001 | < .001 | (205; 60%), (137; 40%) | .947–.954 | < .001 |
| | 3 | 3506.393 | 3675.125 | 3535.547 | .835 | < .001 | < .001 | (51; 15%), (192; 56%), (99; 29%) | .912–.950 | < .001 |
| | **4** | **3352.126** | **3578.380** | **3391.219** | **.844** | **< .01** | **< .01** | **(53; 15%), (149; 44%), (98; 29%), (42; 12%)** | **.902–.927** | **< .001** |
| | 5 | 3284.904 | 3568.680 | 3333.936 | .804 | .31 | .32 | (108; 32%), (55; 16%), (42; 12%), (84; 25%), (53; 15%) | .840–.950 | < .001 |
| GRE | 1 | 5381.876 | 5438.507 | 5394.080 | | | | 422 | | - |
| | 2 | 4366.453 | 4483.758 | 4391.731 | .861 | < .001 | < .001 | (212; 50%), (210; 50%) | .950–.976 | < .001 |
| | 3 | 4147.153 | 4325.133 | 4185.507 | .807 | < .01 | < .01 | (168; 40%), (144; 34%), (110; 26%) | .898–.943 | < .001 |
| | **4** | **3963.036** | **4201.691** | **4014.465** | **.788** | **< .05** | **< .05** | **(86; 20%), (94; 22%), (135; 32%), (107; 25%)** | **.840–.931** | **< .001** |
| | 5 | 3893.506 | 4192.836 | 3958.010 | .810 | .30 | .30 | (75; 18%), (108; 26%), (22; 5%), (108; 26%), (109; 26%) | .827–.940 | < .001 |
| SWI | 1 | 4665.888 | 4720.332 | 4675.916 | | | | 361 | | |
| | 2 | 3699.099 | 3811.877 | 3719.873 | .862 | < .001 | < .001 | (127; 35%), (234; 65%) | .961–.964 | < .001 |
| | **3** | **3498.413** | **3669.524** | **3529.932** | **.812** | **< .05** | **< .05** | **(155; 43%), (76; 21%), (130; 36%)** | **.902–.944** | **< .001** |
| | 4 | 3389.258 | 3618.702 | 3431.522 | .812 | .11 | .11 | (77; 21%), (124; 34%), (71; 20%), (89; 25%) | .864–.940 | < .001 |
| | 5 | 3308.371 | 3596.148 | 3361.381 | .824 | .39 | .39 | (80; 22%), (120; 33%), (67; 19%), (48; 13%), (46; 13%) | .859–.951 | < .001 |

*Note*. GER = Germany; GRE = Greece; SWI = Switzerland; AIC = Akaike information criterion; BIC = Bayesian information criterion; ABIC = sample-size adjusted BIC; LMR LR = Vuong–Lo–Mendell–Rubin Likelihood Ratio Test; ALMR LR = Lo–Mendell–Rubin Adjusted LRT Test; BLRT = bootstrap likelihood ratio test.

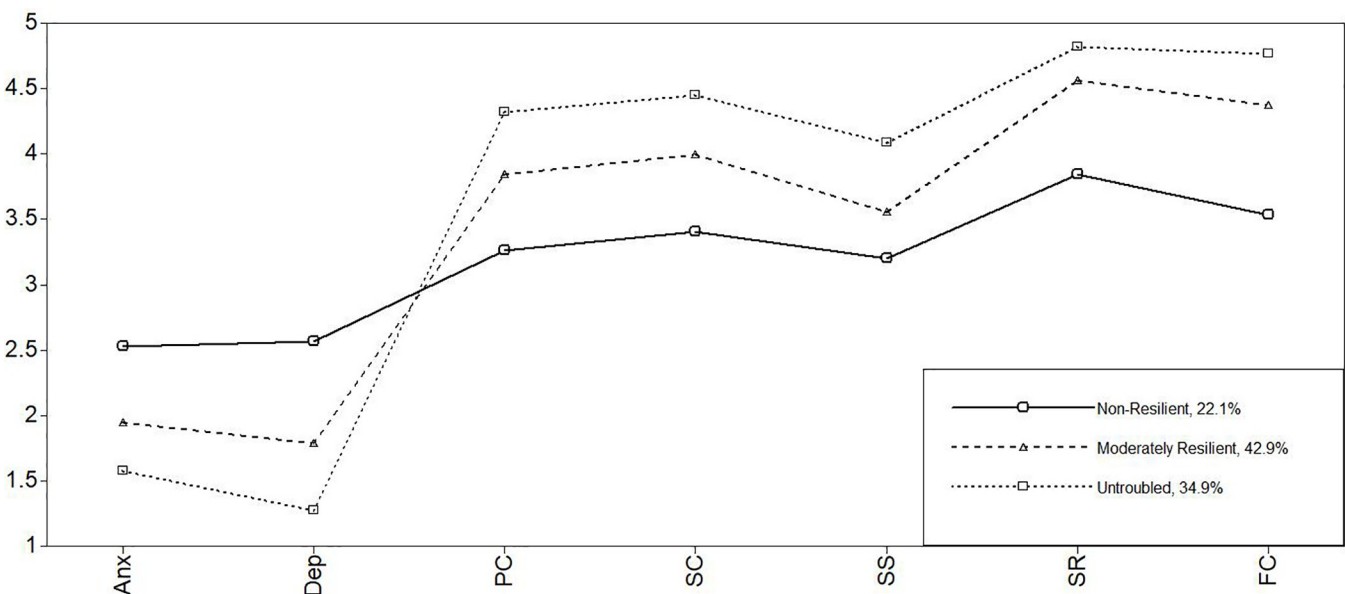

**Fig 1. Latent profiles of Switzerland.** *Note.* Anx = anxiety; Dep = depression; PC = personal competence; SC = social competence; SS = structured style; SR = social resources; FC = family cohesion.

Decreases in the AIC, BIC, and aBIC values from a two-profile to a three-profile solution supported a three-profile solution. Alternatively, in a comparison of the LPA models with one, two, and three profiles, a clear improvement in model fit was found when we moved from one profile to two profiles and from two profiles to three profiles, but a reduced model fit was found when we moved from three profiles to four profiles. The three-profile solution was the most parsimonious model with a reasonable representation of the data and thus was selected for the Swiss subsample. Previous findings also support a three-profile solution [e.g., 69]. Fig 1 displays the profile plot, and Table 2 shows the model results with its parameters. The first profile in the three-profile solution shows a group of students with high symptom levels and low protective factors. They can be considered the nonresilient students (22.1%). The second profile displays a group with moderate symptoms and protective factors. Therefore, we call them the moderately resilient students, who make up about 42.9% of all participants in the Swiss data. Finally, group three shows a profile of low depression and high protective factors—the so called untroubled group (34.9%). A more precise description of each profile follows in the next chapter for all subsamples (including significant differences between the profiles within each model).

For the German and the Greek subsample, a three-profile solution would also be a good solution according to the significant LMRT LR and aLMR LR test, featuring lower AIC, BIC, and aBIC values compared with the two-profile solution. However, when we compare three profiles with four profiles in the German subsample, we see a clear improvement in model fit when checking for the fit indices. All information criteria decreased when we moved from a three- to a four-profile solution. The German sample's entropy value even increased from .835 to .844, and a significant LMR LR test and aLMR LR test, coupled with a satisfying classification accuracy, supported the four-profile solution. The four-profile solution proved to be the most parsimonious model with the most reasonable representation of the data for the German data (see Table 2 and Fig 2).

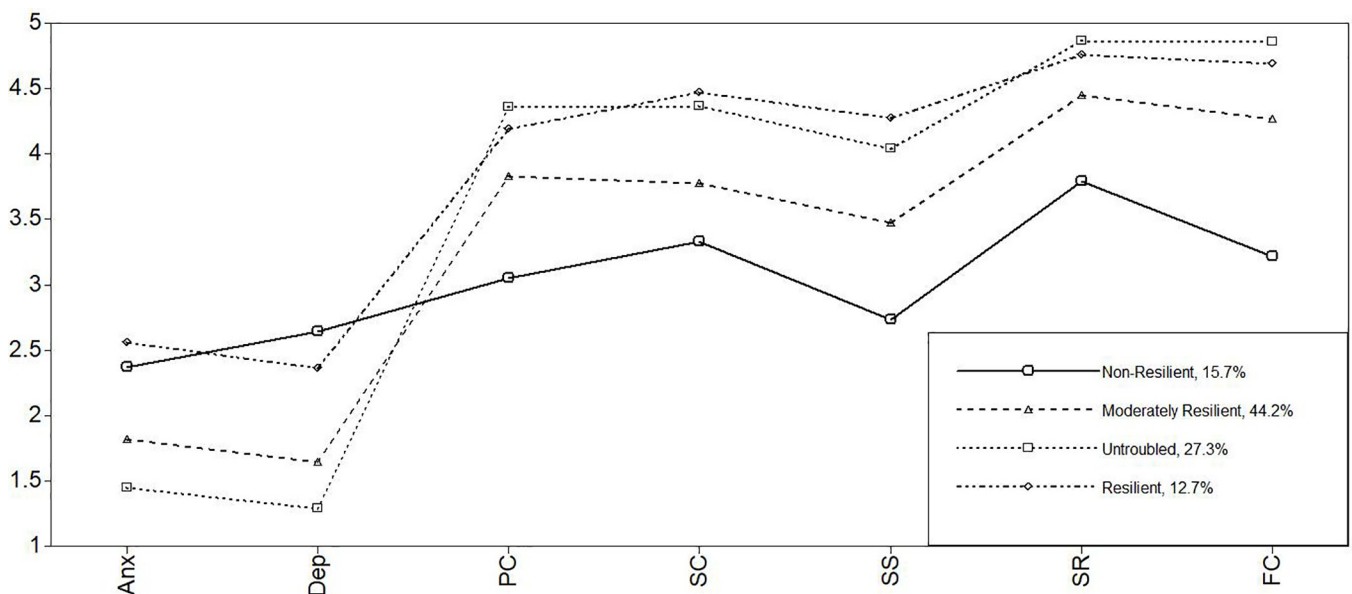

**Fig 2. Latent profiles of Germany.** *Note.* Anx = anxiety; Dep = depression; PC = personal competence; SC = social competence; SS = structured style; SR = social resources; FC = family cohesion.

Similarly to the German subsample, the Greek data indicated the best fit with a four-profile solution. The only apparent difference was the lower entropy value of .788. However, all other fit indices supported the four-profile solution, and again, the most parsimonious model was chosen. Furthermore, studies by Moore et al. [79] and Reinhardt et al. [82] support a four-profile solution (see Table 2 and Fig 3).

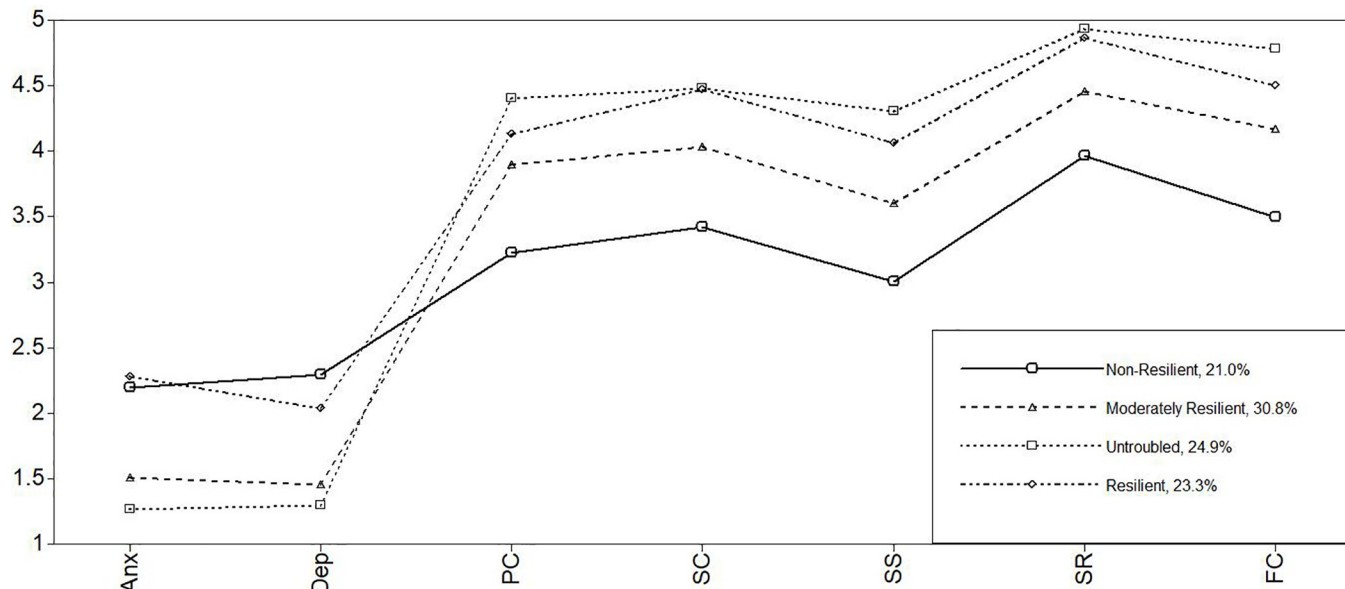

**Fig 3. Latent profiles of Greece.** *Note.* Anx = anxiety; Dep = depression; PC = personal competence; SC = social competence; SS = structured style; SR = social resources; FC = family cohesion.

**Table 3. Wald test, means, and standard errors of the profiles.**

| Variable | Country | Wald's Test | NonResilient[1] | Moderately Resilient[2] | Untroubled[3] | Resilient[4] |
|---|---|---|---|---|---|---|
| | | $\chi 2$ (df) | M (SE) | M (SE) | M (SE) | M (SE) |
| Anxiety | GER | 149.631 (3); $p < .001$ | 2.37 (.13)[2, 3] | 1.81 (.06)[1, 3, 4] | 1.45 (.04)[1, 2, 4] | 2.55 (.13)[2, 3] |
| | GRE | 203.641 (3); $p < .001$ | 2.20 (.11)[2, 3] | 1.51 (.06)[1, 3, 4] | 1.27 (.04)[1, 2, 4] | 2.28 (.13)[2, 3] |
| | SWI | 53.200 (2); $p < .001$ | 2.53 (.14)[2, 3] | 1.95 (.07)[1, 3] | 1.57 (.05)[1, 2] | |
| Depression | GER | 150.779 (3); $p < .001$ | 2.64 (.19)[2, 3] | 1.64 (.07)[1, 3, 4] | 1.29 (.04)[1, 2, 4] | 2.36 (.12)[2, 3] |
| | GRE | 179.145 (3); $p < .001$ | 2.29 (.11)[2, 3] | 1.45 (.04)[1, 3, 4] | 1.30 (.04)[1, 2, 4] | 2.03 (.11)[2, 3] |
| | SWI | 108.334 (2); $p < .001$ | 2.57 (.17)[2, 3] | 1.79 (.06)[1, 3] | 1.27 (.04)[1, 2] | |
| Personal Competence | GER | 105.276 (3); $p < .001$ | 3.05 (.14)[2, 3, 4] | 3.83 (.06)[1, 3, 4] | 4.36 (.06)[1, 2] | 4.19 (.07)[1, 2] |
| | GRE | 171.532 (3); $p < .001$ | 3.23 (.08)[2, 3, 4] | 3.90 (.09)[1, 3, 4] | 4.40 (.05)[1, 2, 4] | 4.13 (.06)[1, 2, 3] |
| | SWI | 123.995 (2); $p < .001$ | 3.26 (.09)[2, 3] | 3.84 (.06)[1, 3] | 4.32 (.06)[1, 2] | |
| Social Competence | GER | 114.931 (3); $p < .001$ | 3.33 (.14)[2, 3, 4] | 3.78 (.07)[1, 3, 4] | 4.37 (.06)[1, 2] | 4.47 (.09)[1, 2] |
| | GRE | 132.138 (3); $p < .001$ | 3.41 (.10)[2, 3, 4] | 4.03 (.10)[1, 3, 4] | 4.48 (.06)[1, 2] | 4.47 (.08)[1, 2] |
| | SWI | 105.972 (2); $p < .001$ | 3.41 (.10)[2, 3] | 3.99 (.07)[1, 3] | 4.45 (.05)[1, 2] | |
| Structured Style | GER | 144.143 (3); $p < .001$ | 2.73 (.14)[2, 3, 4] | 3.47 (.12)[1, 3, 4] | 4.04 (.09)[1, 2, 4] | 4.27 (.07)[1, 2, 3] |
| | GRE | 128.998 (3); $p < .001$ | 3.00 (.11)[2, 3, 4] | 3.60 (.12)[1, 3] | 4.31 (.07)[1, 2] | 4.06 (.10)[1, 2] |
| | SWI | 85.629 (2); $p < .001$ | 3.20 (.10)[2, 3] | 3.55 (.09)[1, 3] | 4.09 (.06)[1, 2] | |
| Social Resources | GER | 91.551 (3); $p < .001$ | 3.79 (.20)[2, 3, 4] | 4.44 (.05)[1, 3, 4] | 4.87 (.03)[1, 2] | 4.76 (.07)[1, 2] |
| | GRE | 110.088 (3); $p < .001$ | 3.97 (.11)[2, 3, 4] | 4.46 (.10)[1, 3, 4] | 4.93 (.02)[1, 2] | 4.86 (.05)[1, 2] |
| | SWI | 84.922 (2); $p < .001$ | 3.84 (.11)[2, 3] | 4.56 (.07)[1, 3] | 4.82 (.04)[1, 2] | |
| Family Cohesion | GER | 165.835 (3); $p < .001$ | 3.21 (.19)[2, 3, 4] | 4.27 (.06)[1, 3, 4] | 4.85 (.03)[1, 2, 4] | 4.69 (.05)[1, 2, 3] |
| | GRE | 149.402 (3); $p < .001$ | 3.49 (.11)[2, 3, 4] | 4.17 (.10)[1, 3, 4] | 4.78 (.04)[1, 2, 4] | 4.50 (.05)[1, 2, 3] |
| | SWI | 101.499 (2); $p < .001$ | 3.53 (.13)[2, 3] | 4.37 (.06)[1, 3] | 4.76 (.03)[1, 2] | |

*Note*. GER = Germany; GRE = Greece; SWI = Switzerland

[1, 2, 3, 4] indicate significant Wald Test.

For the German as well as the Greek data, the four profiles resulted in a very similar picture. Thus, the following labels account for both models. The first profile in the German model (15.7%) and in the Greek model (21.0%), with high symptoms and low protective factors, represent the nonresilient students. The second profile in the German sample (12.7%) and in the Greek sample (23.3%) show high symptoms and protective factors. Thus, they are classified as the resilient profiles. The third profile in the German analysis (44.2%) and in the Greek analysis (30.8%) are the moderately resilient adolescents given their relatively moderate symptoms and protective factors. Finally, the untroubled with high protective factors but very low symptom levels make up profile four in the German (27.3%) and Greek data (24.9%).

**3.2.1 Description and comparison of LPA profiles.** A Wald test revealed an overall significance of the German $\chi^2$ (21) = 763.425, $p < .001$, the Greek $\chi^2$ (21) = 761.069, $p < .001$, and the Swiss model $\chi^2$ (14) = 338.853, $p < .001$, meaning that the profiles in each model are generally different from one another. All pairwise comparisons are displayed in Table 3.

In all three models (Table 3; Figs 1–3), the nonresilient clusters are above the symptoms' mean values (between 2.20 and 2.64) and below the protective factors' mean values (between 2.73 and 3.97) of each country sample (Table 1). Additionally, the nonresilient students' scores are lower in all models, mainly in the areas of personal competence (between 3.05 and 3.26) and structured style (between 2.73 and 3.20).

In the German and Greek models, the nonresilient and the resilient ones have very similar symptom levels (between 2.03 and 2.55). No significant differences exist between the mean values of these two clusters. Any other pairwise comparison with the moderately resilient and

untroubled regarding symptoms is significant. However, they differ clearly in their protective factors. Whereas the nonresilient students reported below-average levels of protective factors, resilient adolescents had the highest protective factor levels aside from the untroubled (between 4.06 and 4.86). Additionally, the resilient and untroubled groups' protective factor levels do not vary much and are relatively similar. Whereas the resilient adolescents clearly show high levels of protective factors, moderately resilient students rank between the lowest and the higher levels of protective factors and symptoms. This group reported average levels of symptoms (between 1.45 and 1.95) and protective factors (between 3.47 and 4.56), similar to the mean values of each country sample.

Interestingly, in all models, the untroubled have by far the lowest level of symptoms (between 1.27 and 1.57) and high protective factors (between 3.47 and 4.56), particularly in the areas of social resources and family cohesion.

In general, the profile plots of the single profiles are quite similar in all country models, leading to the question whether these models and profiles might be directly comparable to each other.

### 3.3 Measurement invariance across countries

In the first MI-LPA analyses, the results from the LRT test rejected measurement invariance in the three-profile model solutions ($\chi2 = 87.03$, $df = 42$, $p < .001$). As such, the three profiles across the countries are not comparable.

In the second MI-LPA analyses using the four-profile model solutions, the results from the LRT test also rejected comparable model solutions across the country samples, as the LRT was significant, ($\chi2 = 114.47$, $df = 56$); $p < .001$).

In the third MI-LPA analyses (Germany and Greece), the results from the LRT test rejected measurement invariance in the four-profile model solutions ($\chi2 = 87.03$, df = 42, $p < .001$). All results of the MI analyses are displayed in Table 4. As such, the four profiles across the German and Greek subsamples are not comparable. Partial measurement invariance could not be established either, as all of the LRT tests resulted in noninvariance.

### 3.4 Predictors of profile membership

Table 5 presents the result from using the R3STEP approach with gender and migration background as predictors of class membership. Pairwise comparisons switching the classes around are displayed accordingly.

For the German data, females were more likely to be in the resilient or moderately resilient profiles compared with the nonresilient profile. Furthermore, the odds of being in the resilient

**Table 4. Measurement invariance across countries.**

| | Model | AIC | BIC | ABIC | Number of free parameters | $H_0$ | $H_0$ scaling correction factor | $\chi^2$ (df) | $p$ | Decision |
|---|---|---|---|---|---|---|---|---|---|---|
| 3 profiles GER, GRE, SWI | unconstrained | 14,839.776 | 15,292.074 | 15,006.209 | 90 | -7329.888 | 1.68 | 87.03 (42) | $p < .001$ | reject |
| | constrained | 14,898.722 | 15,139.948 | 14,987.486 | 48 | -7401.361 | 1.71 | | | |
| 4 profiles GER, GRE, SWI | unconstrained | 14,476.357 | 15,044.243 | 14,685.323 | 113 | -7125.178 | 1.83 | 114.47 (56) | $p < .001$ | reject |
| | constrained | 14,555.627 | 14,842.083 | 14,661.035 | 57 | -7220.814 | 1.99 | | | |
| 4 profiles GER, GRE | unconstrained | 9256.617 | 9613.786 | 9369.278 | 77 | -4551.308 | 1.57 | 117.87 (28) | $p < .001$ | reject |
| | constrained | 9330.046 | 9557.336 | 9401.740 | 49 | -4616.023 | 1.84 | | | |

*Note.* GER = Germany; GRE = Greece; SWI = Switzerland; AIC = Akaike information criterion; BIC = Bayesian information criterion; ABIC = Sample-size adjusted BIC.

**Table 5. Multinomial logistic regression model results for gender and migration background for Germany, Greece, and Switzerland (R3STEP).**

| | Predictor | Nonresilient vs. Moderately Resilient | | Nonresilient vs. Untroubled | | Nonresilient vs. Resilient | | Moderately Resilient vs. Untroubled | | Moderately Resilient vs. Resilient | | Untroubled vs. Resilient | |
|---|---|---|---|---|---|---|---|---|---|---|---|---|---|
| | | Estimate (*SE*) | OR | Estimate (*SE*) | OR | Estimate (*SE*) | OR | Estimate (*SE*) | OR | Estimate (*SE*) | OR | Estimate (*SE*) | OR |
| Germany | Gender[1] | -1.139 ** (0.394) | 0.320 | -0.335 (0.467) | 0.715 | -1.330 ** (0.391) | 0.265 | 0.804 (0.418) | 2.233 | -0.191 (0.318) | 1.211 | -0.995 * (0.406) | 2.704 |
| | Migration Background[2] | -0.200 (0.573) | 0.818 | 0.112 (0.647) | 1.118 | 1.129 * (0.492) | 3.094 | 0.312 (0.636) | 1.366 | -1.330 ** (0.459) | 0.265 | 1.018 (0.545) | 0.361 |
| Greece | Gender[1] | -0.360 (0.362) | 1.433 | -0.987 * (0.382) | 0.373 | -0.235 (0.360) | 0.791 | -0.627 (0.348) | 0.534 | 0.125 (0.328) | 1.133 | 0.752 * (0.332) | 2.122 |
| | Migration Background[2] | -0.911 * (0.360) | 2.487 | -0.380 (0.367) | 0.684 | 0.269 (0.361) | 1.309 | 0.531 (0.342) | 1.700 | 1.180 *** (0.337) | 3.255 | 0.649 (0.333) | 1.914 |
| Switzerland | Gender[1] | 0.108 (0.329) | 1.114 | 0.183 (0.284) | 1.201 | | | 0.075 (0.317) | 1.078 | | | | |
| | Migration Background[2] | -0.117 (0.391) | 0.890 | 0.130 (0.323) | 1.139 | | | 0.247 (0.372) | 1.281 | | | | |

*Note*. Estimate = $\beta$ from R3STEP analysis

*** $p < .001$

** $p < .01$

* $p < .05$.

[1]: 1 = male, 2 = female.

[2]: 0 = migrant, 1 = native.

group compared with the untroubled profile increased by 0.995 (*SE* = 0.406, *OR* = 2.704, *p* < .05) for the girls. Additionally, natives (participants without migration backgrounds) were more likely to be in the nonresilient group compared with the resilient profile. Finally, the probability of natives being in the moderately resilient compared with the resilient group was increased by 1.330 (*SE* = 0.459, *OR* = 0.265, *p* < 0.01). Any other pairwise comparisons in the German sample were nonsignificant.

In the Greek sample, females were less likely to be part of the nonresilient group compared with the untroubled by -0.987 (*SE* = 0.382, *OR* = 0.373, *p* < .05), but they were more likely to belong to the untroubled compared with the resilient. For migration background, significant pairwise differences could be found for comparing the moderately resilient and nonresilient profiles, as well as comparing the moderately resilient with the resilient group. The likelihood of being a part of the moderately resilient group compared with the nonresilient one increased by 0.911 (*SE* = 0.369, *OR* = 2.487, *p* < .05) for natives. Similarly, natives were more likely to be part of the moderately resilient group compared with the resilient group. Any other pairwise comparisons in the Greek sample were nonsignificant.

Finally, for the Swiss profiles, no significant profile differences regarding gender and migration background could be detected. Apparently, gender and migration background did not predict the profiles in the Swiss model in this analysis.

## 4 Discussion

This study explored the resilience profiles of German, Greek, and Swiss adolescents based on the symptoms of anxiety and depression, as well as protective factors in the subdimensions of personal competence, social competence, structured style, social resources, and family cohesion. Subsequently, measurement invariance for a three- and a four-profile solution across all

samples and across the German and Greek model was tested. Finally, the two predictors of gender and migration background were investigated for the final models.

A three- and a four-profile solution support recent previous findings examining resilience and mental health profiles, including at least an above-average, average, and below-average group [e.g., 69, 79, 82, 84, 87]. In the German and Greek sample, a four-profile solution consisting of a nonresilient (i.e., high symptoms and low protective factors), moderately resilient (i.e., moderate symptoms and moderate protective factors), untroubled (i.e., low symptoms and high protective factors), and resilient (i.e., high symptoms and high protective factors) profile were found. Meanwhile, a three-profile solution fit best with the Swiss data, consisting of a nonresilient, moderately resilient, and untroubled groups. Consistent with our expectations, the LPA identified at least three subgroups in the samples studied (hypothesis 1) and resulted in heterogeneity in terms of the resilience outcomes and protective factors, showing two profile solutions for the countries studied. The results indicated that subgroups displaying higher scores for psychological symptoms did not necessarily have lower scores for protective factors, and vice versa. Apparently, nonresilient adolescents did show this pattern of having high anxiety and depression levels while having below-average protective factor scores. Interestingly, the number of students belonging to the nonresilient profile differed highly among the three countries (Germany: 15.7%; Greece: 21.0%; Switzerland: 22.1%). Based on previous research, we expected to find the highest rate in the Greek sample and similarly low rates in Germany and Switzerland [9]. However, when we checked the mean values for symptoms, the Greek sample already showed significantly lower anxiety and HSCL total levels compared with the German and Swiss sample. Furthermore, the resilient profiles were in accordance with the theoretical and empirical backgrounds of resilience, which posit that a certain amount of adversity is necessary to show resilience [30, 69, 77, 78]. Due to the cross-sectional nature of the data, it is not clear whether the number of mental health symptoms, which is just as high as in the nonresilient subgroup, is adequate for successful development over time. Therefore, further investigation is needed with longitudinal data. However, clear differences can be found between the country samples. Almost double the number of Greek adolescents were part of the resilient profile (23.3%) compared with the German profile (12.7%).

Not very surprisingly, the moderately resilient students reported symptoms and protective factor levels just around the mean scores of the three country samples and represented the largest subgroups (Germany: 44.2%; Greece: 30.8%; Switzerland: 42.9%). It would be very interesting to see how this group develops over time because a shift toward any other group is certainly possible according to resilience theory [30].

In the German and Greek profiles, the group size of the untroubled was similar (Germany: 27.3%; Greece: 24.9%), whereas the largest group of untroubled individuals was represented in the Swiss data (34.9%).

In general, these findings are in line with the results of the KIDSCREEN study, where the majority was considered to be among the normal healthy groups [9].

Even though we chose the best-fitting four-profile solution for the German and Greek data, model fit values were acceptable for a three-profile solution too. It is not obvious why the German and Greek models are more nuanced and differ from the Swiss model with only three profiles. However, it is possible that the resilient profile in the German and Greek models are absorbed in different profiles. When investigating the distributions of the profiles in each country and comparing them, we can see that the untroubled group is clearly smaller in both four-profile models compared to the Swiss model. The resilient group shares high levels of protective factors comparable to the untroubled ones. Furthermore, there are less nonresilient students in the German sample that have similar levels of symptoms compared to the resilient group, whereas less pupils are part of the moderately resilient group in the Greek sample. In

the moderately resilient profile of the Greek model, protective factor levels are closer to the resilient and untroubled group compared to the symptom levels. Therefore, it is possible that more adolescents were absorbed from the moderately resilient group to the resilient group in the Greek model.

Furthermore, we can only assume structural influences might be involved, which would have to be further investigated. Switzerland is not part of the European Union (EU). The field of EU policy covers not only foreign and security policy but also education, training and youth, human rights and democracy, and culture [115]. The EU supports member states in their efforts to improve the quality and efficiency of education, resulting in the so-called 11 European Youth Goals. Besides fostering quality learning, the equality of all genders, mental health and wellbeing, and inclusive societies are part of these goals [116]. This leads to the assumption that a common goal, framework, and structure might influence the findings of this present study, as Switzerland is not part of the EU. The primary responsibility for education resides with the 26 cantons of Switzerland unless the federal constitution stipulates that the confederation be in charge [117]. However, with the Intercantonal Agreement on Harmonisation of Compulsory Education (HarmoS Agreement, [118]), the curriculum and its most important objectives were harmonized nationwide. This could result in a more homogeneous response patterns and thus a lower number of profiles. Furthermore, the distance between the schools taking part in the data collection is shorter in Switzerland than it is in Germany and especially in Greece, which could result in more homogeneous response patterns in these two countries. It would be very interesting to check for cantonal differences within the Swiss data but also replicate the German and Greek results with similar samples in other/closer country regions.

Measurement invariance did not support comparable profiles, as neither a three-profile solution across all subsamples nor a four-profile solution for the German and Greek subsamples was supported (hypothesis 2). Therefore, these profiles are not directly comparable, even when they show very similar patterns. Consequently, the implications for theory and praxis need to be sample specific, supporting the idea of resilience being a cultural- and contextual-sensitive concept [1, 4, 54, 55].

However, at this point, it is crucial to deeply discuss bias in cross-cultural studies. It is well known that these studies are majorly challenged by the validity and applicability of instruments. Three types of biases are relevant for international studies: construct bias, method bias, and item bias [119]. First, construct bias is possible if definitions, construct-relevant aspects, or behaviors vary across cultures [120]. One cannot rule out the fact that symptoms and protective factors could be understood differently among all students. No studies on the measurement invariance of the instruments exist across these three countries. "Cultural conventions about the self, reality, social rules, and patterns of emotional expression, for example, simply make universal criteria of psychiatric illness difficult to attain and the idea itself problematical" [121]. Second, method bias includes sample bias, instrument bias, and administration bias. When confounding variables (e.g., education level), characteristics, or the application of the measure (e.g., language differences) differ clearly, samples might not be directly comparable [122]. Third, item bias is concerned with the different meanings of items across cultures. For avoiding bias as much as possible, validated scales have been used. The HSCL [97] has been validated in German and in Greek [123]. The German version of the READ [33] has been cross-culturally validated across a German and a Swiss sample [96]. Unfortunately, no validated Greek version for the READ exists. However, the striking difference found between the models in this study is between the Swiss and the German/Greek model, not between the two models where a fully validated questionnaire was used. Furthermore, all three samples

included seventh graders in rural and urban schools, and administrators followed a protocol to avoid administration bias while collecting the data.

Finally, the predictors of gender and migration background were included in the analysis (hypothesis 3). Gender and migration background were not associated with the profiles in the Swiss model. It is not a very big surprise, considering the Global Gender Gap Index (including economic participation and opportunity, educational attainment, health and survival, and political empowerment) from 2021, which ranks Switzerland 10 among 156 countries [124]. Nevertheless, the findings of these studies regarding gender differences can be a surprise when one checks for educational attainment and health survival in the Global Gender Gap Index, where Switzerland is ranked at 80 out of 128 countries [124]. Similarly, no association between migration background and the profiles leads to more assumptions that are ambiguous. On the one hand, having a migration background might be not relevant due to the fact that approximately 50% of all participating students have one. On the other hand, according to a review of migrants in Switzerland, psychological problems more frequently affect children with migration backgrounds, leaving them with distinct health needs [125]. Therefore, we are very cautious about the present findings.

Unlike in the Swiss model, gender and migration background were associated with the profiles in the German and Greek samples. However, the results from these countries differed even though the profiles of the LPA were quite similar. In the German sample, girls were more likely to be in the nonresilient profile than in the other profiles, and they were also more likely to be in the untroubled profile compared with the resilient profile. When consulting the Global Gender Gap Repot Index again, we can expect gender differences due to the ranking. Germany ranks 11th overall, places 55th in educational attainment, and 75th in health survival [124]. However, it is very interesting that if girls are showing symptoms of anxiety and stress, it is much more likely that they do not show adequate protective factors, which is in line with previously mentioned findings. Family cohesion is a strong protective factor for girls [40, 41], and protective factors against depressive symptoms are particularly strong for girls [45]. As we can see in Fig 2, depressive symptoms are extremely high, whereas family cohesion is relatively low in the nonresilient profile compared with the other profiles. This also supports the fact that girls are more likely to be part of the untroubled profile compared with the resilient profile. Family cohesion levels are higher, and depression levels are the lowest. Furthermore, migrants were more likely to be in the resilient profile compared with the nonresilient, but they were less likely to be in the resilient profile when compared with the moderately resilient profile. Apparently, the untroubled profile is not associated with migration background, which could be traced back to the greater challenges that students with a migration background have to face. Thus, it is not surprising that they are instead part of the other profiles. However, it seems that if adolescents with migration backgrounds are dealing satisfactorily with challenges, they are in the moderately resilient or resilient profiles. However, the levels of emotional suffering should not be underestimated. The results also indicate that students with migration backgrounds are more likely to deal with certain levels of symptoms, as also indicated by a recent study investigating mental health among immigrant adolescents in Germany. Thus, they should be supported in dealing with distress [126].

For the Greek sample, the opposite was true. Boys were more likely to be in the resilient profile compared with the untroubled profile. Furthermore, girls were less likely to be part of the nonresilient profile compared with the untroubled. Again, finding gender differences when one consults the Global Gender Report Index is not surprising. Greek ranks 98th overall, 107th in health survival, and 67th in educational attainment [124]. Boys show higher levels of symptoms than girls do but deal well with them, whereas girls have high levels of protective factors and low symptom levels. As mentioned earlier, the KIDSCREEN study also detected

more boys in the "abnormal" profile with psychological distress compared with girls [9]. Finally, having a migration background was associated with an increased probability of being a part of the nonresilient or resilient profile compared with the moderately resilient profile. Thus, having a migration background is not associated with low levels of symptoms (similarly to the adolescents in Germany). Instead, it is more likely that it shows a high level of symptoms, and the ability or inability to deal well with these symptoms determines whether one is part of the nonresilient or resilient profile. Because Greece has been a transit and host country for immigrants for many decades, and because it has more recently been transformed into a host country for refugees from countries of war, it is not surprising that migration background is associated with profiles containing high levels of symptoms [127]. However, it would be very interesting to differentiate between migrants and refugees who are dealing with different challenges in life, particularly in a transit country, such as Greece.

## 4.1 Conclusion for theory and practice

Based on the results of the present study, we should rethink the cut-off values of symptoms and protective factors (very likely of risk factors, too). We use universal values instead of making allowances for cultural and contextual specifications. Neither adolescents among different cultures, nor girls and boys nor having a migration background or not can be considered to be one totality and one universal group. This also may indicate that interventions and preventions could be differentiated depending on the profiles of the target groups/samples. When grouping people, we should always keep in mind that within-group differences indicate different needs of the individuals. For interventions at schools, (individual) protective factors and a good school climate might be individually fostered to provide opportunities to develop and improve these protective factors in a safe environment. School climate is also one of the most significant predictors regarding adolescents' psychological adjustment in general [128]. Furthermore, the person-centered approach creates a typology of real individuals and thus fosters an understanding of their profiles based on symptom variables and protective factor variables. Instead of representing a group of children in one general model, the results highlight the importance of individual differences. Protective factors or symptoms should not be overinterpreted, which can result in stereotypes. It is crucial to keep in mind that no "model child" exists, and that showing high symptoms does not mean the child does not have any protective factors, or vice versa.

Finally, we have an urgent need for more longitudinal studies across various cultural and contextual groups that cover more indicators and different combinations of indicators of resilience to support the idea of resilience (adaptation over time) much more.

## 4.2 Limitations and suggestions for further research directions

Several study limitations need to be addressed. First, although the sample size for each analysis was acceptable [129], it was a very specific sample (seventh graders [ISCED 2]) that has not been further investigated. We do not have any further background information on possibly relevant variables such as socioeconomic status or any other family background variables, even though it is well known that the family strongly influence symptoms and protective factors [e.g., 130]. Furthermore, it was a sample of a generally healthy population. Additional research should test these models in populations with, for example, adolescent psychiatric patients. It would also be of great interest to replicate the analysis in a general population of adults and children or a similar adolescent sample in another country or region.

Second, resilience profiles should ideally include at least one risk factor, one protective factor, and one outcome variable to elucidate the ameliorative effects of the protective factor at

high- and low-risk conditions or the moderation of the effect of the risk factor on negative out-comes. Thus, it would be helpful to include variables such as stressful life events as risk factors in future studies.

Third, and probably most importantly, speaking of resilience should include a longitudinal perspective and risk factors, too. We were able to present only a picture of a modeled reality at a certain point in time based on two outcome variables and protective factors in a cross-sectional design. Thus, these results relate to only a momentary understanding of profiles, not a resilience pattern across time. In the future, similar analyses should be conducted where individual development can be recorded. The adjustment according to a risk factor provides much more insight into someone's resilience than investigating only symptoms and protective factors (and risk factors) simultaneously.

Fourth, we called the low symptoms and high protective factors profile the untroubled profile. This might be a bit misleading because we covered only certain specific aspects of resilience profiles (i.e., depressive and anxiety symptoms, as well as the protective factors of the READ). However, it is very likely that other symptoms and/or risks that we did not investigate affected these students as well. Therefore, we do not conclude that these adolescents are completely untroubled and risk/symptom free. They are, compared with the other profiles, untroubled based only on the analysis of the present indicators.

Fifth, even though the READ has been validated for the German-speaking version [96], no validation of the Greek version has taken place. All other (sub)scales are reliable and valid scales for measuring symptoms and protective factors. A Greek validation would strengthen the analysis of the Greek data and their results.

Sixth, migration background was coded as a binary variable (0; 1). This clearly reduces the validity of the present study and should therefore be investigated again by using a more distinctive and elaborate coding system. It is absolutely possible that findings, particularly concerning the Swiss model, would differ.

Finally, examining the predictive value of gender and of migration background could be questionable considering that resilience is a group-specific construct. Measurement invariance across groups (such as gender and migration background) should be conducted to check for the comparability of these groups. It is possible that females and males would show significantly different profiles when compared separately within each country sample and across the entire sample. The same applies for migration background. Therefore, the results concerning group membership should be interpreted with caution.

## Supporting information

**S1 File. Dataset.**
(SAV)

## Acknowledgments

The authors thank all participating schools, teachers, and students involved in the study in all three countries. We also highly appreciate the valuable contributions of all student assistants who helped during data collection.

## Author Contributions

**Conceptualization:** Clarissa Janousch, Frederick Anyan, Wassilis Kassis, Roxanna Morote, Odin Hjemdal.

**Data curation:** Clarissa Janousch.

**Formal analysis:** Clarissa Janousch.

**Funding acquisition:** Wassilis Kassis.

**Investigation:** Clarissa Janousch.

**Methodology:** Clarissa Janousch.

**Supervision:** Wassilis Kassis.

**Visualization:** Clarissa Janousch.

**Writing – original draft:** Clarissa Janousch.

**Writing – review & editing:** Clarissa Janousch, Frederick Anyan, Wassilis Kassis, Roxanna Morote, Odin Hjemdal, Petra Sidler, Ulrike Graf, Christian Rietz, Raia Chouvati, Christos Govaris.

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
