## [Decision Letter · Decision Letter 0]

6 Dec 2021

PONE-D-21-24447

Resilience profiles across context: a latent profile analysis in a German, Greek, and Swiss sample

PLOS ONE

Dear Dr. Janousch,

Thank you for submitting your manuscript to PLOS ONE. After careful consideration, we feel that it has merit but does not fully meet PLOS ONE’s publication criteria as it currently stands. Therefore, we invite you to submit a revised version of the manuscript that addresses the points raised during the review process.

We look forward to receiving your revised manuscript.

Kind regards,

Meng-Cheng Wang

Academic Editor

PLOS ONE

“The authors thank all participating schools, teachers, and students involved in the study in all three countries and are grateful for the financial support by the Swiss National Science Foundation (SNSF) and the University of Applied Sciences and Arts Northwestern Switzerland (FHNW).”

“This study was funded by the Swiss National Science Foundation (SNSF) through the National Centre of Competence in Research (NCCR) – on the move via the project Overcoming Inequalities with Education – School Resilience, grant number 51NF40-182897, awarded to WK. Additionally, the University of Applied Sciences and Arts Northwestern Switzerland (FHNW) generously supported the study. The funders had no role in study design, data collection and analysis, decision to publish, or preparation of the manuscript.”

Reviewers' comments:

Reviewer's Responses to Questions

**Comments to the Author**

1. Is the manuscript technically sound, and do the data support the conclusions?

Reviewer #1: Yes

Reviewer #2: Partly

2. Has the statistical analysis been performed appropriately and rigorously? 

Reviewer #1: Yes

Reviewer #2: Yes

3. Have the authors made all data underlying the findings in their manuscript fully available?

Reviewer #1: Yes

Reviewer #2: Yes

4. Is the manuscript presented in an intelligible fashion and written in standard English?

Reviewer #1: Yes

Reviewer #2: Yes

5. Review Comments to the Author

Reviewer #1: Dear editor,

Thank you for the opportunity to review this fine and interesting paper. The article reports the process of using LPA to test Resilience profiles across context in a German, Greek, and Swiss sample. The manuscript is well written throughout, stating with a good, exhaustive, and balanced introduction and ending with an adequate discussion of the study’s findings and limitations. Below I specify some minor issues that could help to improve the manuscript’s strength.

Abstract

1.I suggest authors change “Measurement invariance did not hold” to “Measurement invariance did not hold across three countries…”.

2. I suggest add more information about the students (N, M, SD) and profiles (N, proportion%, et al) in abstract section.

Introduction

Introduction is well written and informative. However, I have some minor remarks:

3.p.2. line 26-28 – I suggest authors add more references in this paragraph, and more evidence need to be presented to clarify that why conduct this research.

4.p.3. line 40-49 – Authors introduced a cross-cultural study in detail, “These findings can be explained by several…” I suggest authors added 1-2 sentence(s) to introduce how several factors (the families’ socioeconomic situation or structural aspects, parental mental health issues…) affect the proportion difference in results across countries.

5. I strongly suggested authors added new meta-analysis or review studies to include some information about why it is important to contains these variables in this study.

Person-Centered Approaches in Resilience Research

6. Author introduce well why they used LPA to investigate resilience in this section. However, I have some minor issues. First, I suggested each studies introduced must include resilience profiles’ number, size, proportion et al. Which country they conducted, they used which assessment tools. I thought these were important because this might lead to the difference in LPA results. Besides, author argue that gender and migration might be important, I suggest add related information in these studies.

Statistical Analyses

7.p.11. line 269 – MI is necessary when comparing various groups, but why MI across gender and migration were not tested in the present study? I suggest present it in the 3.3 section.

Discussion

8. p.20. line 423-438 Its very interesting that only Switzerland sample could yield a 3-profile solution but four in Germany and Greece sample although in nearly the same sample size. I suggest author also present several sentences to clarify 3-profile solution of Germany and Greece sample, I assumed that Resilient profile might be absorbed in certain profile (perhaps untroubled profile). I thought the 3-profile solution plots might be similar across three countries. I suggest this should be discussed more in this section.

Reviewer #2: Thanks for inviting me to review this paper, entitled “Resilience profiles across context: a latent profile analysis in a German, Greek, and Swiss sample”. Generally, this study is well organized with a clear research question, comprehensive literature review and technically sound analyses. It shows that the patterns of resilience profiles are different in different cultures: a three-profile-solution for Switzerland, and a four-profile solution found for Germany and Greece. The findings shed lights on the question “Which promotive and protective factors or processes are best for which people in which contexts at what level of risk exposure and for which outcomes? ”. However, I also have some minor concerns as follows:

1. I noted that the participants in the current study are all adolescents aged between 11- and 16-year-old, not including anyone with other age. Given that, it is better for you to focus on that specific group and limit the subject of the current study to adolescents since the results may not necessarily be generalized to other groups with other age (e.g., adults).

2. I feel a little bit confused how you measured resilience and what you based on to conduct a LPA (the basis for categorization). I thought resilience was measured with READ, including personnel competence, social competence, structured style, social resources, and family cohesion. But I noticed that you reported the comparisons results across resilience profiles on anxiety, depression, as well as the five protective factors measured with READ (in Table 3). If you measured resilience with READ and categorized the sample based on that, why did you then compare the determined profiles resulting from the five factors on these five factors?

3. I agree that the limitations you highlighted are much needed to improve this study. Apart from that, I am interested in why you considered the three cultural contexts: German, Greece, and Switzerland. Because of convenient sampling or other reasons?

4. Please pay more attention to the formatting rules. For example, the punctuation sign may be lost after the phrases of “in their three classes” at the end of page 6 (Line 156).

Hope these comments are helpful for you.

6. PLOS authors have the option to publish the peer review history of their article (what does this mean?). If published, this will include your full peer review and any attached files.

Reviewer #1: No

Reviewer #2: No

---

## [Author Response · Author response to Decision Letter 0]

6 Jan 2022

Response to Reviewers

Manuscript ID: PONE-D-21-24447

Title: "Resilience profiles across context: a latent profile analysis in a German, Greek, and Swiss sample" 

Revised title: "Resilience profiles across context: a latent profile analysis in a German, Greek, and Swiss sample of adolescents"

Editor: Dr. Meng-Cheng Wang

Authors: 

Dear Dr. Meng-Cheng Wang

Thank you very much for your comments and the invitation to submit a revised version of the manuscript. Your feedback as well as the reviewers’ feedback is greatly appreciated and helpful. In the meantime, we have revised the manuscript based on all the comments and concerns and hope that it is now acceptable for publication in PLOS ONE.

Editor:

Authors: 

Our sincerest apologies for any mistakes in the formatting style. Thank you for the templates. We have changed the authors’ affiliations according to the template. Furthermore, the submitted files are labelled according to the requirements as Response to Reviewers, Revised Manuscript with Track Changes and Manuscript. Additionally, the headings have been adapted to 18pt./16pt./14pt, only the first words of the headings have been capitalized, and figures have been labelled as Fig(s) (throughout the whole Revised Manuscript with Track Changes and the Manuscript documents).

Editor: 

“The authors thank all participating schools, teachers, and students involved in the study in all three countries and are grateful for the financial support by the Swiss National Science Foundation (SNSF) and the University of Applied Sciences and Arts Northwestern Switzerland (FHNW).”

“This study was funded by the Swiss National Science Foundation (SNSF) through the National Centre of Competence in Research (NCCR) – on the move via the project Overcoming Inequalities with Education – School Resilience, grant number 51NF40-182897, awarded to WK. Additionally, the University of Applied Sciences and Arts Northwestern Switzerland (FHNW) generously supported the study. The funders had no role in study design, data collection and analysis, decision to publish, or preparation of the manuscript.”

Authors: 

Thank you for changing the online submission form on our behalf. We would like to include the following amended statements. The acknowledgement statement has also been changed in the manuscript:

Revised Manuscript, p. 27, l. 616-617:

“Acknowledgements

The authors thank all participating schools, teachers, and students involved in the study in all three countries. We also highly appreciate the valuable contributions of all student assistants who helped during data collection.” 

We hope that the funding section is fine without any changes since we have removed any funding-related text from the manuscript.

“Funding

This study was funded by the Swiss National Science Foundation (SNSF) through the National Centre of Competence in Research (NCCR) – on the move via the project Overcoming Inequalities with Education – School Resilience, grant number 51NF40-182897, awarded to WK. Additionally, the University of Applied Sciences and Arts Northwestern Switzerland (FHNW) generously supported the study. The funders had no role in study design, data collection and analysis, decision to publish, or preparation of the manuscript.”

Editor: 

Authors: 

We have added the subsection Supporting information at the end of the manuscript (S1 File. Dataset.). Additionally, we have included a note in the Methods section (at the beginning of Participants paragraph where the data is mentioned): 

Revised Manuscript, p. 9, l. 230-231:

“This study examined data (data available in the Supporting information section, S1 File) collected as part of the National Centres of Competence in Research (NCCR) […].”

Revised Manuscript, p. 27, l. 619:

“S1 File. Dataset.”

Editor: 

Authors: 

We have not cited papers that have been retracted. However, we have made changes to the reference list because of the changes to the manuscript suggested by the Reviewers. We have added the following references throughout the manuscript:

Revised Manuscript, p. 27ff, l. 621-940:

1. Ungar M, editor. Multisystemic resilience: adaptation and transformation in contexts of change. New York: Oxford University Press; 2021. 

2. Quinlan AE, Berbés-Blázquez M, Haider LJ, Peterson GD. Measuring and assessing resilience: broadening understanding through multiple disciplinary perspectives. Allen C, editor. J Appl Ecol. 2016 Jun;53(3):677–87. 

3. Panter-Brick C. Culture and Resilience: Next Steps for Theory and Practice. In: Theron LC, Liebenberg L, Ungar M, editors. Youth Resilience and Culture [Internet]. Dordrecht: Springer Netherlands; 2015 [cited 2021 Dec 10]. p. 233–44. (Cross-Cultural Advancements in Positive Psychology; vol. 11). Available from: http://link.springer.com/10.1007/978-94-017-9415-2_17

49. Windle G, Bennett KM, Noyes J. A methodological review of resilience measurement scales. Health Qual Life Outcomes. 2011;9(1):8.

50. Lee JH, Nam SK, Kim A-R, Kim B, Lee MY, Lee SM. Resilience: A Meta-Analytic Approach. Journal of Counseling & Development. 2013 Jul;91(3):269–79.

51. Hu T, Zhang D, Wang J. A meta-analysis of the trait resilience and mental health. Personality and Individual Differences. 2015 Apr;76:18–27.

70. Seery MD, Holman EA, Silver RC. Whatever does not kill us: Cumulative lifetime adversity, vulnerability, and resilience. Journal of Personality and Social Psychology. 2010 Dec;99(6):1025–41. 

71. Seery MD, Leo RJ, Holman AE, Silver RC. Lifetime exposure to adversity predicts functional impairment and healthcare utilization among individuals with chronic back pain. Pain. 2010 Sep;150(3):507–15. 

72. Smith BW, Dalen J, Wiggins K, Tooley E, Christopher P, Bernard J. The brief resilience scale: Assessing the ability to bounce back. Int J Behav Med. 2008 Sep;15(3):194–200. 

73. Snyder CR, Harris C, Anderson JR, Holleran SA, Irving LM, Sigmon ST, et al. The will and the ways: Development and validation of an individual-differences measure of hope. Journal of Personality and Social Psychology. 1991;60(4):570–85. 

74. Chen G, Gully SM, Eden D. Validation of a New General Self-Efficacy Scale. Organizational Research Methods. 2001 Jan;4(1):62–83. 

75. Bell BS, Kozlowski SWJ. A Typology of Virtual Teams: Implications for Effective Leadership. Group & Organization Management. 2002 Mar;27(1):14–49. 

76. Scheier MF, Carver CS, Bridges MW. Distinguishing optimism from neuroticism (and trait anxiety, self-mastery, and self-esteem): A reevaluation of the Life Orientation Test. Journal of Personality and Social Psychology. 1994;67(6):1063–78. 

80. You S, Furlong MJ, Dowdy E, Renshaw TL, Smith DC, O’Malley MD. Further Validation of the Social and Emotional Health Survey for High School Students. Applied Research Quality Life. 2014 Dec;9(4):997–1015. 

81. Goodman R. The Strengths and Difficulties Questionnaire: A Research Note. J Child Psychol & Psychiat. 1997 Jul;38(5):581–6. 

83. Keyes CLM. Mental health in adolescence: Is America’s youth flourishing? American Journal of Orthopsychiatry. 2006 Jul;76(3):395–402.

85. Odenstad A, Hjern A, Lindblad F, Rasmussen F, Vinnerljung B, Dalen M. Does age at adoption and geographic origin matter? A national cohort study of cognitive test performance in adult inter-country adoptees. Psychol Med. 2008 Dec;38(12):1803–14.

86. Zimet GD, Dahlem NW, Zimet SG, Farley GK. The Multidimensional Scale of Perceived Social Support. Journal of Personality Assessment. 1988 Mar;52(1):30–41.

88. Lehman AF. A quality of life interview for the chronically mentally ill. Evaluation and Program Planning. 1988 Jan;11(1):51–62. 

89. Derogatis LR. BSI brief symptom inventory: Administration, scoring, and procedures manual (4th ed.). Minneapolis, MN: National Computer Systems; 1993. 

90. McLellan AT, Kushner H, Metzger D, Peters R, Smith I, Grissom G, et al. The fifth edition of the addiction severity index. Journal of Substance Abuse Treatment. 1992 Jun;9(3):199–213. 

91. Wagnild GM, Young H. Development and psychometric evaluation of the Resilience Scale. Journal of Nursing Measurement. 1993;1(2):165–78. 

92. Garnefski N, Kraaij V. Cognitive emotion regulation questionnaire – development of a short 18-item version (CERQ-short). Personality and Individual Differences. 2006 Oct;41(6):1045–53. 

 

Reviewers' comments:

Reviewer's Responses to Questions

Comments to the Author

1. Is the manuscript technically sound, and do the data support the conclusions?

Reviewer #1: Yes

Reviewer #2: Partly

2. Has the statistical analysis been performed appropriately and rigorously? 

Reviewer #1: Yes

Reviewer #2: Yes

3. Have the authors made all data underlying the findings in their manuscript fully available?

Reviewer #1: Yes

Reviewer #2: Yes

4. Is the manuscript presented in an intelligible fashion and written in standard English?

Reviewer #1: Yes

Reviewer #2: Yes

 

5. Review Comments to the Author

Reviewer #1: Dear editor,

Thank you for the opportunity to review this fine and interesting paper. The article reports the process of using LPA to test Resilience profiles across context in a German, Greek, and Swiss sample. The manuscript is well written throughout, stating with a good, exhaustive, and balanced introduction and ending with an adequate discussion of the study’s findings and limitations. Below I specify some minor issues that could help to improve the manuscript’s strength.

Authors: 

Dear reviewer, many thanks for your insightful and much-appreciated feedback that you have provided. We have revised the manuscript in response to your helpful comments and concerns. 

Abstract

1.I suggest authors change “Measurement invariance did not hold” to “Measurement invariance did not hold across three countries…”.

Authors: 

Thank you for this suggestion. We have made changes accordingly in the abstract section.

Revised Manuscript, p. 2, l. 22/23:

“Measurement invariance did not hold across the three countries.”

Reviewer 1: 

2. I suggest add more information about the students (N, M, SD) and profiles (N, proportion%, et al) in abstract section.

Authors: 

Thank you also for this suggestion. We have added this information in the abstract section.

Revised Manuscript, p. 2, l. 15-22:

“The present study investigated resilience profiles (based on levels of symptoms of anxiety and depression and five dimensions of protective factors) of 1,160 students from Germany (n = 346, 46.0% females, Mage = 12.77, SDage = 0.78), Greece (n = 439, 54.5% females, Mage = 12.68, SDage = 0.69), and Switzerland (n = 375, 44.5% females, Mage = 12.29, SDage = 0.88) using latent profile analyses. We also checked for measurement invariance and investigated the influence of gender and migration on class membership. A three-profile-solution was found for Switzerland (nonresilient 22.1%, moderately resilient 42.9%, untroubled 34.9%), and a four-profile-solution was the best fitting model for Germany (nonresilient 15.7%, moderately resilient 44.2%, untroubled 27.3%, resilient 12.7%) and Greece (nonresilient 21.0%, moderately resilient 30.8%, untroubled 24.9%, resilient 23.3%).”

Reviewer 1: 

Introduction

Introduction is well written and informative. However, I have some minor remarks:

3.p.2. line 26-28 – I suggest authors add more references in this paragraph, and more evidence need to be presented to clarify that why conduct this research.

Authors: 

We agree that we can extend this paragraph even though we wanted to keep it rather general in the beginning. Therefore, we have added references and more evidence:

Revised Manuscript, p. 2, l. 29-35: 

“Instead of only understanding resilience as a linear set of causal relationships, recent research on resilience has focused on the multisystemic aspect of the concept (1). Resilience of human and ecological systems are mutually dependent on each other (2) and therefore, resilience needs to be studied by taking the different contexts of these systems and their connectivity into account. Furthermore, resilience is a normative concept that is highly influenced by cultural aspects such as moral values, and structural and social dimensions (3). Ungar and Theron raised the question, “Which promotive and protective factors or processes are best for which people in which contexts at what level of risk exposure and for which outcomes? (4)” 

Reviewer 1: 

4. p.3. line 40-49 – Authors introduced a cross-cultural study in detail, “These findings can be explained by several…” I suggest authors added 1-2 sentence(s) to introduce how several factors (the families’ socioeconomic situation or structural aspects, parental mental health issues…) affect the proportion difference in results across countries.

Authors: 

Thank you very much for this feedback. We fully agree that because of the paper’s focus this important aspect needs to be explained a bit more in detail. Therefore, we have added the following sentences:

Revised Manuscript, p. 3, l. 53-59:

“These findings can be explained by several factors, such as the families’ respective socioeconomic situation (which might be influenced by the economic situation of a country or region) that has proven to be associated via perceived social status with adolescents’ mental health (10). Additionally, structural aspects, such as parental mental health issues, domestic violence, or poor peer support can influence adolescents’ mental health (9,11). All these aspects are unique but closely linked factors contributing to adolescents’ mental health and resilience. However, the extent of influence of each factor differs due to cultural features in a society.”

Reviewer 1: 

5. I strongly suggested authors added new meta-analysis or review studies to include some information about why it is important to contain these variables in this study.

Authors: 

We have added the following paragraph to support the approach we have chosen in the present study:

Revised Manuscript, p. 5, l. 93-107:

“A methodological review of resilience scales has already shown in 2011 (49) that measuring resilience is challenging due to its ambiguous definition. Most of the time, resilience is being measured as the presence or absence of assets and resources that facilitate resilience as a process, but no “gold standard” was found among the 15 measures. Even though, resilience scales mostly cover only assets and resources, models and concepts of resilience go beyond analyzing these aspects and include risks factors, often focusing on the absence of negative indicators of mental health (e.g., anxiety or depression). A meta-analysis of 31,071 participants in 33 studies investigated the relationship between psychological resilience and relevant variables (50). All selected articles stem from the years 2001 to 2010 and results indicated as expected that protective factors, such as self-efficacy, life satisfaction, or optimism have the biggest effect on resilience. In addition, medium effects were measured for risk factors, such as depression, anxiety, or PTSD. Finally, demographic variables contributed small effects, but were still important to resilience. Another meta-analysis supported these findings by analyzing the relation between resilience, mental health, and demographics in 60 studies representing 68,720 participants (51). High correlations were found between resilience and mental health. Additionally, gender was moderating this relationship. More attention needs to be paid to females experiencing higher levels of mental health problems and lower levels of protective factors. Therefore, when investigating resilience, it is crucial to not only consider the protective and risk factors but also to include mental health aspects and demographics in the research questions.” 

Reviewer 1: 

Person-Centered Approaches in Resilience Research

6. Author introduce well why they used LPA to investigate resilience in this section. However, I have some minor issues. First, I suggested each studies introduced must include resilience profiles’ number, size, proportion et al. Which country they conducted, they used which assessment tools. I thought these were important because this might lead to the difference in LPA results. Besides, author argue that gender and migration might be important, I suggest add related information in these studies.

Authors: 

Thank you for your suggestions. We agree that these aspects might lead to different LPA/LCA results. Therefore, we included all information available regarding profile numbers, proportions, country information, assessment tools, and gender/migration information of introduced studies:

Revised Manuscript, p. 6, l. 147-210:

“There is an increasing number of empirical studies examining profiles (aspects) of resilience using latent class and profile analyses (LCA & LPA). A recent study from the United Kingdom and Western Australia, for instance, focused mainly on adversity (i.e., different configurations of lifetime adversities) and resilience resources (i.e., bounce-back, hope, self-efficacy, and optimism) (69). The team used the adapted version of the cumulative lifetime adversity measure (70,71), the Bounce-back ability (72), the Adult Hope (73), the General Self-Efficacy (74) / a self-efficacy scale by Bell and Kozlowski (75), and the Life Orientation Test-Revised Scale (76). They conducted two separate studies with a general (N = 1,506, 48.2% females) and a university sample (N = 348, 61.5% females). Results revealed three profiles for each sample showing statistically different levels of resilience in the three detected classes. For the general sample, the biggest group was the Moderate class (62.7%) followed by the High (20.5%) and Low (16.8%) Polyadversity classes. In the university sample, the Low (41.1%) and High Polyadversity (41.1%) classes showed identical group sizes, while the Vicarious Adversity (17.8%) was the smallest group. Differences between all three latent classes in both subsamples in terms of individual-level resilience resources were mixed. Individuals in the Moderate Polyadversity class reported the highest level of resilience in the general population study. These findings were statistically significant when comparing the Moderate class with the High Polyadversity class, but only partially significant (for bounce-back resilience and optimism) in comparing the Moderate with the Low Polyadversity classes. According to Lines et al. (69) and previous studies (77,78), a moderate amount of exposure to adversity is ideal for opportunities to develop protective factors, and therefore to support resilience. 

Furthermore, being less exposed to adversity led to more resilience resources (protective factors) when comparing the Low and High Polyadversity classes. Being exposed to a high amount of adversity is highly detrimental regarding the availability of protective factors. Additionally, being exposed to fewer adversities might give fewer opportunities to develop necessary resilience resources compared to a moderate amount of adversities. Additionally, gender differences were detected in both samples. Females were more likely than males to be part of the High Polyadversity class than the Vicarious Adversity or Low Polyadversity classes in the university sample, while males were more likely to be part of the Low Polyadversity class compared to the Moderate Polyadversity class in the general sample. No information was given on migration background.

Another study focusing on mental health classifications examined profiles of American high school students (N = 332, 48.5% females) over 3 years using a dual-factor construct of mental health (79). Measurements used included the Social-Emotional Health Survey-Secondary (80) and the Strengths and Difficulties Questionnaire (81). Independent LPAs for each grade (9–12) based on four positive mental health domains and internalizing and externalizing problems revealed four distinct subgroups— Complete Mental Health, Moderately Mentally Healthy, Symptomatic but Content, and Troubled. Like the general population sample in the study mentioned above (69), most students were in the Complete (30.5% Grade 9, 40.8% Grade 10, 20.5% Grade 11) or Moderately Mentally Healthy (43.4%, 32.0%, 44.3%) classes. The Troubled class (5.7%, 6.0%, 3.8%) represented the smallest number of individuals across all grades, while the Symptomatic but Content class (20.3%, 21.2%, 31.3%) was between these classes. Higher levels of distress and lower levels of strength were reportedly associated with fewer symptoms of anxiety and depression. No further investigations on gender and migration background were made. 

This four-profile-solution was confirmed by Reinhardt et al. (82) based on three well-being indicators (emotional, psychological, and social aspects), resulting in the Languishing, Moderate Mental Health, Emotionally Vulnerable, and Flourishing classes. 1,572 (51% females) Hungarian adolescents filled out a questionnaire including the Adolescent Mental Health Continuum – Short Form (83) and the Strength and Difficulties Questionnaire (81). 39% were part of the Moderate Mental Health group, 11% belonged to the Emotionally Vulnerable class. The Languishing class, including 14% of the sample, reported low levels of prosocial behavior, high rates of peer problems, and loneliness. In contrast, lower levels of loneliness, more prosocial behavior, and fewer emotional problems and peer problems predicted the Flourishing class (36%) in comparison to the Languishing class. Furthermore, the Flourishing category included more males and younger adolescents compared to the Languishing group. No further gender differences were reported, nor was any information on migration background. 

Two further recent studies focused on risk and protective factors. Mohanty et al. (84) were able to demonstrate that protective factors on all levels might play a crucial role in preventing the occurrence of risks in their three classes: Moderate (39.5%), Protective (34.3%), and High-risk (26.2%). In the study, 953 (67.2% females) participants answered a questionnaire including two items measuring pre-adoption risk (85), eleven self-created items about post-adoption risk (84), the Multidimensional Scale of Perceived Social Support (86), and a single item asking “how many close friends do you have?”. Findings suggest that social support in particular ameliorated negative effects of risks. More males were part of the Moderate class, while more females were part of the Protective and High-risk classes. Migration background has not been investigated. 

Finally, a four-class-solution is supported by Altena et al. (87), confirming that accumulated protective factors are important in preserving a certain quality of life. Findings resulted in the four classes, High-Risk and Least Protected (24%), Higher Functioning and Protected (14%), At-Risk (45%), and Low-Risk (17%) classes. 251 adolescents (32% females) participated in the study that were asked as a single item whether they have been abused. Furthermore, they answered questions from the Lehman Quality of Life Interview (88), the Brief Symptom Inventory-53 (89), the European Addiction Severity Index (90), the Resilience Scale (91) and the Cognitive Emotion Regulation Questionnaire (92). No gender differences existed between the subgroups and migration background was not investigated in the study.

However, both studies included a very specific sample. Mohanty et al. (84) focused on Korean adult international adoptees, whereas Altena et al. (87) investigated homeless young adults in the Netherlands. Thus, findings need to be treated with caution when comparing to more general samples.” 

Reviewer 1: 

Statistical Analyses

7.p.11. line 269 – MI is necessary when comparing various groups, but why MI across gender and migration were not tested in the present study? I suggest present it in the 3.3 section.

Authors: 

Thank you for this very interesting point. We fully agree and acknowledge your concern that MI across gender and migration background are also important and should be further investigated. Therefore, we had it already mentioned in the limitations section. However, the aim of this paper was giving a first insight into the country-differences and using gender and migration background solely as predictive variables. We believe that investigating MI across gender and migration background within each country sample and across the entire sample would result in completely new research questions, which is beyond the scope of the present paper. Such questions may be:

(1.1) How many resilience profiles based on symptoms (depression and anxiety) and protective factors (personal competence, social competence, structured style, social resources, and family cohesion) can be found for females and males across the entire sample?

(1.2) How many resilience profiles based on symptoms (depression and anxiety) and protective factors (personal competence, social competence, structured style, social resources, and family cohesion) can be found for females and males within each country-sample?

(1.3) How many resilience profiles based on symptoms (depression and anxiety) and protective factors (personal competence, social competence, structured style, social resources, and family cohesion) can be found for migrants and natives across the entire sample?

(1.4) How many resilience profiles based on symptoms (depression and anxiety) and protective factors (personal competence, social competence, structured style, social resources, and family cohesion) can be found for migrants and natives within each country-sample?

 (2.1) Do identical resilience profiles exist across gender across the entire sample?

(2.2) Do identical resilience profiles exist across gender within each country-sample?

(2.3) Do identical resilience profiles exist across migration background across the entire sample?

(2.4) Do identical resilience profiles exist across migration background within each country-sample?

(3) Are e.g. socioeconomic status and age predictors of these latent resilience profiles?

etc.

Additionally, the samples are quite small for further distinction between females/males and migrants/natives within the country-samples. The distribution for each profile would shrink to approximately half the sample size. Nevertheless, we are also very interested in these findings but we would prefer to run these analyses separately and rigorously – if even possible. Still, we have expanded on what we already had mentioned in the limitations section:

Revised Manuscript, p. 27, l. 609-614:

“Finally, examining the predictive value of gender and of migration background could be questionable considering that resilience is a group-specific construct. Measurement invariance across groups (such as gender and migration background) should be conducted to check for the comparability of these groups. It is possible that females and males would show significantly different profiles when compared separately within each country sample and across the entire sample. The same applies for migration background. Therefore, the results concerning group membership should be interpreted with caution.” 

Reviewer 1: 

Discussion

8. p.20. line 423-438 Its very interesting that only Switzerland sample could yield a 3-profile solution but four in Germany and Greece sample although in nearly the same sample size. I suggest author also present several sentences to clarify 3-profile solution of Germany and Greece sample, I assumed that Resilient profile might be absorbed in certain profile (perhaps untroubled profile). I thought the 3-profile solution plots might be similar across three countries. I suggest this should be discussed more in this section.

Authors:

Thank you very much for this highly appreciated input. We agree that the resilient profile has been absorbed in certain profiles, probably mainly in the untroubled profile. However, it is also very likely that the resilient profile has been absorbed in the nonresilient profile in Germany and in the moderately resilient profile in Greece.

Therefore, we have added several sentences in this paragraph, clarifying the 3/4-profile solution of Germany and Greece.

Revised Manuscript, p. 22, l. 469-479:

“Even though we chose the best-fitting four-profile solution for the German and Greek data, model fit values were acceptable for a three-profile solution too. It is not obvious why the German and Greek models are more nuanced and differ from the Swiss model with only three profiles. However, it is possible that the resilient profile in the German and Greek models are absorbed in different profiles. When investigating the distributions of the profiles in each country and comparing them, we can see that the untroubled group is clearly smaller in both four-profile models compared to the Swiss model. The resilient group shares high levels of protective factors comparable to the untroubled ones. Furthermore, there are less nonresilient students in the German sample that have similar levels of symptoms compared to the resilient group, whereas less pupils are part of the moderately resilient group in the Greek sample. In the moderately resilient profile of the Greek model, protective factor levels are closer to the resilient and untroubled group compared to the symptom levels. Therefore, it is possible that more adolescents were absorbed from the moderately resilient group to the resilient group in the Greek model.”

Authors: 

Again, we would like to thank you for your helpful and much appreciated feedback! We hope that by considering all of your supportive feedback, we have increased the quality of the paper significantly. 

 

Reviewer #2: Thanks for inviting me to review this paper, entitled “Resilience profiles across context: a latent profile analysis in a German, Greek, and Swiss sample”. Generally, this study is well organized with a clear research question, comprehensive literature review and technically sound analyses. It shows that the patterns of resilience profiles are different in different cultures: a three-profile-solution for Switzerland, and a four-profile solution found for Germany and Greece. The findings shed lights on the question “Which promotive and protective factors or processes are best for which people in which contexts at what level of risk exposure and for which outcomes?”. However, I also have some minor concerns as follows:

Reviewer 2: 

1. I noted that the participants in the current study are all adolescents aged between 11- and 16-year-old, not including anyone with other age. Given that, it is better for you to focus on that specific group and limit the subject of the current study to adolescents since the results may not necessarily be generalized to other groups with other age (e.g., adults).

Authors:

Thank you very much for this input. We completely agree, and have changed the title accordingly to show that our focus is solely on adolescents.

Revised Manuscript, p. 1, l. 0:

“Resilience profiles across context: a latent profile analysis in a German, Greek, and Swiss sample of adolescents” 

Reviewer 2:

2. I feel a little bit confused how you measured resilience and what you based on to conduct a LPA (the basis for categorization). I thought resilience was measured with READ, including personnel competence, social competence, structured style, social resources, and family cohesion. But I noticed that you reported the comparisons results across resilience profiles on anxiety, depression, as well as the five protective factors measured with READ (in Table 3). If you measured resilience with READ and categorized the sample based on that, why did you then compare the determined profiles resulting from the five factors on these five factors?

Authors:

We apologize for causing any confusion. 

It is correct that we have not only included the READ into our analyses but also the HSCL-25 (anxiety and depression). The READ only measures protective factors in the five subdimensions (personal competence, social competence, structural style, social resources, and family cohesion). Our resilience model goes beyond investigating solemnly protective factors (that are nevertheless very important) by expanding the model with levels of symptoms. As shown in the analyses, resilience is highly dependent on protective, risk factors and symptoms. Therefore, we have added the symptoms as additional information about the adolescents’ resilience and determined the profiles on these seven indicators. Otherwise, we would have analyzed protective factor profiles with a different resilience measurement model. 

Revised Manuscript, p. 5, l. 108-116:

“For several decades, resilience scientists have been on a prolonged mission to understand mental health issues to prevent and treat them by examining risk and protective factors. Instead of focusing on pathways leading toward psychopathology in pathogenesis, resilience research arose from attempts to account for both positive and negative patterns (52) based on a salutogenic approach to health (53). Despite definitional ambiguity, newer definitions reflect the perspective of resilience as a complex, dynamic, and adaptive system that goes beyond the idea of an individual bouncing back and recovering from a traumatic experience. For the purpose of this paper, we define resilience according to Masten as “the capacity of a dynamic system to adapt successfully to disturbances that threaten systemic function, viability, or development” (30). This definition does not only accentuates the multisystemic nature of resilience but subsequently acknowledges the importance of cultural narratives and contextual realities in mental health and resilience research (1,4,54,55).” 

Reviewer 2: 

3. I agree that the limitations you highlighted are much needed to improve this study. Apart from that, I am interested in why you considered the three cultural contexts: German, Greece, and Switzerland. Because of convenient sampling or other reasons?

Authors:

Thank you very much for giving us the chance to clarify this point. The three countries were included because of the initial aim of the entire research project. As mentioned in the materials and methods section, the data was collected as part of a NCCR – on the move project. All projects funded by the NCCR – on the move lay strong focus on migration and mobility. Since Greece has been a very important country for migrants as a first arrival and transit country, and Germany and Switzerland are common countries where migrants seek permanent residence, these three countries were chosen as part of the project Overcoming Inequalities with Education – School and Resilience. However, the present study does not lay such strong focus on the migration aspects and therefore, data was used for the present research questions. 

Reviewer 2: 

4. Please pay more attention to the formatting rules. For example, the punctuation sign may be lost after the phrases of “in their three classes” at the end of page 6 (Line 156).

Authors: 

Our sincerest apologies for mistakes in the formatting style. We have made changes across the whole manuscript according to the formatting rules of PLOS ONE. All headings, figure labels and document names have been changed. 

Reviewer 2:

Hope these comments are helpful for you.

Authors: 

Indeed, thank you very much for your much appreciated and helpful feedback! We hope that we were able to increase the quality of the present paper by considering all of your supportive feedback. 

 

6. PLOS authors have the option to publish the peer review history of their article (what does this mean?). If published, this will include your full peer review and any attached files.

Do you want your identity to be public for this peer review? For information about this choice, including consent withdrawal, please see our Privacy Policy.

Reviewer #1: No

Reviewer #2: No

---

## [Editor Report · Decision Letter 1]

12 Jan 2022

Resilience profiles across context: a latent profile analysis in a German, Greek, and Swiss sample of adolescents

PONE-D-21-24447R1

Dear Dr. Janousch,

We’re pleased to inform you that your manuscript has been judged scientifically suitable for publication and will be formally accepted for publication once it meets all outstanding technical requirements.

Kind regards,

Meng-Cheng Wang

Academic Editor

PLOS ONE
---

## [Editor Report · Acceptance letter]

14 Jan 2022

PONE-D-21-24447R1 

Resilience profiles across context: a latent profile analysis in a German, Greek, and Swiss sample of adolescents 

Dear Dr. Janousch:

I'm pleased to inform you that your manuscript has been deemed suitable for publication in PLOS ONE. Congratulations! Your manuscript is now with our production department. 

Kind regards, 

on behalf of

Dr. Meng-Cheng Wang 

Academic Editor

PLOS ONE